# pCoMole: Pareto-Constrained Molecule Editing with Discrete Flows

**Tong Chen,**[1,*] **Maximilian Holsman,**[2,*] **Lin Zhao,**[3] **Pranam Chatterjee**[1,3,†]

[1]Department of Computer and Information Science, University of Pennsylvania
[2]Department of Computer Science, Duke University
[3]Department of Bioengineering, University of Pennsylvania

[†]Corresponding author: pranam@seas.upenn.edu

## Abstract

Biomolecular therapeutics often start from known sequences and require targeted editing to improve multiple properties while satisfying hard biochemical and manufacturability constraints. Existing generative methods do not jointly support multi-objective optimization, hard feasibility, and sequence editing in discrete, variable-length biological spaces. We introduce **P**areto-**Co**nstrained **Mol**ecule **e**diting (**pCoMole**), a framework built on discrete flow matching that steers a pre-trained Edit Flow toward user-specified preferences while enforcing terminal feasibility. pCoMole defines a feasibility-gated terminal distribution using an augmented Tchebycheff utility and realizes the resulting preference tilt through a Doob-$h$ transform of the underlying edit process. To make this construction practical, we approximate the required harmonic function using short Monte Carlo rollouts over candidate edits, yielding an efficient guided editor with provable preference consistency. We validate pCoMole by shrinking GFP while retaining fluorescence-related properties, shortening diverse Cas9 orthologs while preserving PAM specificity, and compressing peptide binders into short peptidomimetics that optimize seven drug-related properties under hard constraints. Overall, pCoMole enables constraint-aware, Pareto-aligned editing of biomolecular sequences in discrete, variable-length spaces.

## 1 Introduction

Biomolecular therapeutics are rarely designed from scratch (Packer & Liu, 2015; McLure et al., 2022). Engineering typically proceeds by editing existing sequences to improve efficacy and developability while obeying hard biochemical and manufacturability constraints. In many settings, *sequence shrinkage* is a primary objective: shorter molecules are cheaper to synthesize, easier to manufacture, and more amenable to delivery. This pressure is especially acute for genome editors such as CRISPR-Cas9, whose large size poses a fundamental barrier to viral delivery and *in vivo* translation (Wang et al., 2020; Kabadi et al., 2024), motivating aggressive shrinkage while preserving PAM specificity and catalytic function. Similar edit-based optimization arises across modalities. For insulin analogs, targeted edits tune absorption kinetics and duration while preserving receptor engagement and formulation stability (Howey et al., 1994; Kramer et al., 2021). For monoclonal antibodies such as adalimumab, sequence modifications reduce immunogenicity and aggregation risk without sacrificing affinity or specificity (Weinblatt et al., 2003). For peptide therapeutics such as the revolutionary GLP-1R agonists, optimization explicitly includes shortening or removing dispensable segments to reduce synthesis cost while retaining biological activity (Swedberg et al., 2016; Han et al., 2024; Yoshida et al., 2022). These examples illustrate that biomolecular design is fundamentally an *editing problem*, often driven by shrinkage and requiring multi-objective optimization under hard terminal constraints.

Several recent methods address protein shrinkage directly. RayGun (Devkota et al., 2024) performs template-guided miniaturization by encoding variable-length protein sequences into a fixed-dimensional embedding space and decoding them at shorter lengths, enabling large indels while

broadly preserving structure. SCISOR (Baron et al., 2025) instead formulates shrinkage as a discrete diffusion process trained to plan deletions by reversing an insertion-only noising process, yielding state-of-the-art predictions of deletion effects and improved motif preservation. While effective for shrinkage, these methods are not designed for general multi-objective editing: they do not jointly optimize multiple developability or functional objectives under explicit feasibility constraints, nor do they provide a mechanism for preference-driven trade-off control during editing.

More broadly, constrained multi-objective molecular optimization has been studied using evolutionary search and reinforcement learning (Xia et al., 2025; Lin et al., 2026; Sun et al., 2022; Yang et al., 2024; Olivecrona et al., 2017; You et al., 2018; Zhou et al., 2019; Loeffler et al., 2024). These approaches can approximate Pareto fronts but typically rely on repeated online oracle evaluations and are not naturally suited to offline editing of incumbent molecules in large, variable-length discrete spaces. Discrete diffusion and discrete flow matching models have recently emerged as powerful generators for sequence spaces (Austin et al., 2021; Campbell et al., 2022; Gat et al., 2024), with guidance mechanisms enabling control over objectives and constraints (Li et al., 2022; Schiff et al., 2024; Tang et al., 2025b; Zhang et al., 2025). However, most controllable discrete generators focus on *de novo* generation and address either constraint satisfaction or preference control in isolation, without supporting iterative sequence editing with insertions and deletions (Chen et al., 2025b;a; Tang et al., 2025a;c; Vincoff et al., 2025). Edit Flows extend discrete flow matching to stochastic sequence editing via local insertions, deletions, and substitutions, but have not been developed as a constraint-aware, multi-objective editing framework for biomolecular design (Havasi et al., 2025).

To address this gap, we introduce **P**areto-**Co**nstrained **Mol**ecule **e**diting (**pCoMole** a framework for offline, constraint-aware multi-objective sequence editing. pCoMole steers a pre-trained Edit Flow by defining a feasibility-gated terminal distribution ranked by an augmented Tchebycheff utility, and realizes this preference tilt through a Doob-$h$ transform. Practical guidance is achieved via short Monte Carlo rollouts that approximate the required harmonic function, enabling efficient editing in large, variable-length discrete biomolecular spaces.

**Our key contributions are:**

1. We introduce pCoMole, an offline method for Pareto-aligned biomolecular sequence editing under hard terminal constraints.

2. We derive a Doob-$h$ guided editing process with an augmented Tchebycheff utility and make it practical using short rollout-based guidance with theoretical guarantees.

3. We demonstrate pCoMole across multiple biomolecular editing tasks, including GFP and Cas9 shrinkage and *in silico* peptidomimetic binder compression.

4. We show improved feasibility and trade-off control relative to state-of-the-art shrinkage and editing methods, including RayGun and SCISOR.

A detailed discussion of Related Work is provided in Appendix Section A.

## 2 BACKGROUND

**Continuous-time Markov chains**  Let $\mathcal{X}$ be a discrete state space and $(X_t)_{t \in [0,1]}$ a continuous-time Markov chain (CTMC) on $\mathcal{X}$ with time-inhomogeneous rate $u_t(\cdot \mid x_t)$. The CTMC is characterized by the first-order expansion

$$\Pr(X_{t+h} = x \mid X_t = x_t) = \delta_{x_t}(x) + h\, u_t(x \mid x_t) + o(h), \tag{1}$$

where $\delta_{x_t}(x)$ is the Kronecker delta and $\lim_{h \to 0} o(h)/h = 0$. Valid rates satisfy the rate conditions:

$$u_t(x \mid x_t) \geq 0 \ \ \forall x \neq x_t, \qquad \sum_{x \in \mathcal{X}} u_t(x \mid x_t) = 0, \tag{2}$$

which implies $u_t(x_t \mid x_t) = -\sum_{x \neq x_t} u_t(x \mid x_t)$ so that the time marginals $p_t(x) = \Pr(X_t = x)$ obey the Kolmogorov forward equation.

**Edit Flows.**  Edit Flows define a CTMC directly on the space of variable-length sequences (Havasi et al., 2025). Let $\mathcal{T}$ denote a vocabulary of size $M$ and let $\mathcal{X} = \bigcup_{n=0}^{N} \mathcal{T}^n$ be the set of all sequences

up to length $N$. The model parameterizes a rate field $u_t^\theta(\cdot \mid x)$ whose support is restricted to sequences that differ from $x$ by a single edit. For a sequence $x = (x_1, \ldots, x_{n(x)})$ and position $i \in \{1, \ldots, n(x)\}$, the edit operations are insertion $\mathrm{ins}(x, i, a)$, deletion $\mathrm{del}(x, i)$, and substitution $\mathrm{sub}(x, i, a)$, where $a \in \mathcal{T}$. Since these operations yield mutually exclusive outcomes, the rate for each possible edit is parameterized by a type-specific total rate and, when applicable, a token distribution:

$$u_t^\theta(\mathrm{ins}(x, i, a) \mid x) = \lambda_{t,i}^{\mathrm{ins}}(x)\, Q_{t,i}^{\mathrm{ins}}(a \mid x), \tag{3}$$

$$u_t^\theta(\mathrm{del}(x, i) \mid x) = \lambda_{t,i}^{\mathrm{del}}(x), \tag{4}$$

$$u_t^\theta(\mathrm{sub}(x, i, a) \mid x) = \lambda_{t,i}^{\mathrm{sub}}(x)\, Q_{t,i}^{\mathrm{sub}}(a \mid x). \tag{5}$$

Here $\lambda_{t,i}^{\mathrm{ins}}(x)$, $\lambda_{t,i}^{\mathrm{del}}(x)$, and $\lambda_{t,i}^{\mathrm{sub}}(x)$ are nonnegative total rates that control how frequently each edit type occurs at position $i$, and $Q_{t,i}^{\mathrm{ins}}(\cdot \mid x)$ and $Q_{t,i}^{\mathrm{sub}}(\cdot \mid x)$ are normalized distributions over token values. The diagonal term $u_t^\theta(x \mid x)$ is determined by the rate conditions and equals the negative sum of all insertion, deletion, and substitution total rates across positions, ensuring that the CTMC is well-defined.

Edit Flows are trained using an auxiliary alignment variable to resolve the ambiguity that multiple edit paths can map $x_0$ to $x_1$. The method augments the token space with a blank symbol $\varepsilon$ and considers aligned sequences $z \in \mathcal{Z} = (\mathcal{T} \cup \{\varepsilon\})^N$, together with a projection $f_{\mathrm{frm\text{-}blank}} : \mathcal{Z} \to \mathcal{X}$ that removes blanks. Training constructs pairs $(z_0, z_1)$ that project to $(x_0, x_1)$ and samples intermediate alignments $z_t$ along a token-wise mixture path with schedule $\kappa_t$. Each mismatch coordinate between $z_t$ and $z_1$ induces a unique single-edit update of the projected sequence $x_t = f_{\mathrm{frm\text{-}blank}}(z_t)$, which provides a tractable supervision signal for the Edit Flow rates. The resulting objective penalizes large total outgoing rate while increasing the log-rate assigned to the alignment-implied edit:

$$\mathcal{L}(\theta) = \mathbb{E}\left[ \sum_{x \neq x_t} u_t^\theta(x \mid x_t) - w_t \sum_{i \in \Delta_t} \log u_t^\theta\big(x_t^{(i)} \mid x_t\big) \right], \tag{6}$$

where $w_t = \dot{\kappa}_t / (1 - \kappa_t)$, $\Delta_t = \{\, i : z_t^i \neq z_1^i \,\}$, and $x_t^{(i)}$ denotes the sequence obtained by replacing the $i$-th coordinate of $z_t$ by $z_1^i$ and projecting back via $f_{\mathrm{frm\text{-}blank}}$.

In the rest of the paper, we use $x$ to denote a generic sequence in $\mathcal{X}$ and write $x_t$ for the state of the CTMC at time $t$; all rate functions such as $u_t(\cdot \mid x)$, $\lambda_{t,p}(x)$, and $\pi_{t,p}^{\mathrm{type}}(\cdot \mid x)$ are defined for generic $x$ and are evaluated at $x = x_t$ inside the training objective.

## 3 METHODS

### 3.1 PROBLEM SETUP

Let $\mathcal{T}$ be a finite token vocabulary and let $\mathcal{X} = \bigcup_{n \geq 0} \mathcal{T}^n$ denote the set of variable-length token sequences. We consider $K$ objective functions $f_k : \mathcal{X} \to \mathbb{R}$ for $k \in \{1, \ldots, K\}$ to be maximized, together with $M$ inequality constraints $g_m : \mathcal{X} \to \mathbb{R}$ for $m \in \{1, \ldots, M\}$ and $L$ equality constraints $h_\ell : \mathcal{X} \to \mathbb{R}$ for $\ell \in \{1, \ldots, L\}$. The feasible set is

$$\mathcal{F} := \{x \in \mathcal{X} : g_m(x) \leq 0\, \forall m, \ \ h_\ell(x) = 0\, \forall \ell\}. \tag{7}$$

Given an input sequence $x_0 \in \mathcal{X}$ and a pre-trained Edit Flow that induces a conditional terminal distribution $q(\cdot \mid x_0)$ over edited sequences, our goal is to sample terminal sequences $x_1 \sim q(\cdot \mid x_0)$ that solve a multi-objective optimization problem by improving the objective vector $(f_1(x_1), \ldots, f_K(x_1))$ while satisfying $x_1 \in \mathcal{F}$. Intermediate states visited by the editing process are not required to satisfy the constraints.

### 3.2 PCOMOLE WITH DOOB-$h$ GUIDED EDIT FLOWS

With a pre-trained Edit Flow in hand, we now turn our attention to develop **pCoMole**, a **p**areto-**Co**nstrained **mol**ecule **e**diting algorithm, to edit molecule sequences for optimizing multiple objectives while maintaining feasibility (Figure 1). Let $u_t(\cdot \mid x)$ denote the fixed rate field of the trained Edit

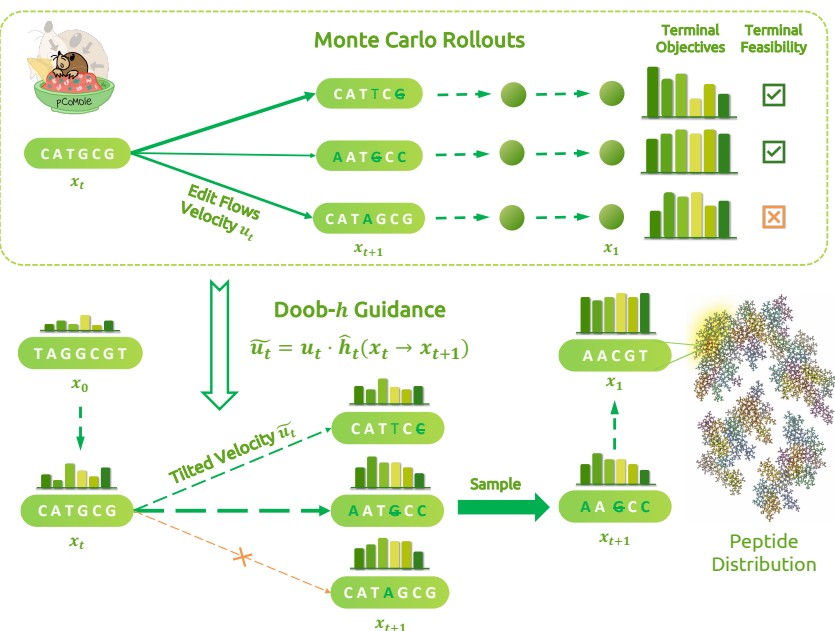

Figure 1: **pCoMole overview.** From the current sequence $x_t$, pCoMole samples candidate edits using the pre-trained Edit Flows velocity $u_t$. Short Monte Carlo rollouts estimate terminal utility and feasibility, yielding an approximate Doob-$h$ function that reweights transitions into a tilted velocity $\tilde{u}_t$. Sampling under $\tilde{u}_t$ steers editing toward feasible, high-utility terminal sequences.

Flow and let $(X_t)_{t\in[0,1]}$ be the induced time-inhomogeneous CTMC with $X_0 = x_0$, which defines a terminal distribution $q(\cdot \mid x_0)$ at $t = 1$. pCoMole biases this terminal distribution toward feasible molecules with improved multi-objective performance by specifying a terminal preference function $G : \mathcal{X} \to \mathbb{R}_{\geq 0}$.

We construct $G$ from an augmented Tchebycheff utility. Let $f(x) = (f_1(x), \ldots, f_K(x))$ be objective scores to be maximized, let $\omega \in \mathbb{R}_{>0}^K$ be a weight vector, and let $r \in \mathbb{R}^K$ be a reference point. We define

$$U(x) := \min_{k\in\{1,\ldots,K\}} \omega_k \left( f_k(x) - r_k \right) + \rho \sum_{k=1}^{K} \omega_k \left( f_k(x) - r_k \right), \tag{8}$$

where $\rho > 0$ is the augmentation coefficient. This utility is strictly monotone with respect to Pareto dominance for $\omega > 0$, and therefore encourages improvements across objectives rather than collapsing to a single criterion. We incorporate feasibility by defining

$$G(x) := \exp\!\left(\beta\, U(x)\right) \mathbb{I}[x \in \mathcal{F}], \tag{9}$$

with inverse-temperature $\beta > 0$. The corresponding target terminal distribution is the tilted law

$$q_G(x_1 \mid x_0) = \frac{q(x_1 \mid x_0)\, G(x_1)}{\sum_{x'} q(x' \mid x_0)\, G(x')} \propto q(x_1 \mid x_0)\, G(x_1), \tag{10}$$

which reweights base Edit Flow samples by preference while assigning zero mass to infeasible terminals.

The Doob-$h$ transform provides a principled way to realize equation 10 through a modified CTMC that preserves the local edit structure of the base process. Define the harmonic function

$$h_t(x) := \mathbb{E}[G(X_1) \mid X_t = x], \tag{11}$$

where the conditional expectation is taken under the base rates $u_t$. The Doob-$h$ guided process is the CTMC with off-diagonal rates

$$u_t^G(y \mid x) := u_t(y \mid x) \frac{h_t(y)}{h_t(x)}, \qquad y \neq x, \tag{12}$$

and diagonal term determined by the rate conditions.

**Proposition 3.1.** *Assume $G(X_1)$ is integrable under the base Edit Flow and $h_t(x) > 0$ for all $(t, x)$ reachable from $x_0$. The CTMC with rates equation 12 has terminal distribution $q_G(\cdot \mid x_0)$ defined in equation 10.*

The proof is provided in Appendix C.1. Proposition 3.1 justifies our design by showing that, in the ideal setting where $h_t$ is available, the guided edit process samples exactly from the preference-tilted terminal distribution that encodes both feasibility and multi-objective optimization through $G$. Varying the weight vector $\omega$ yields different tilts of $q(\cdot \mid x_0)$, which allows pCoMole to explore distinct Pareto trade-offs while remaining anchored to the pre-trained Edit Flow dynamics. The proofs for Pareto optimality and coverage (Proposition C.2) are provided in Appendix C.

### 3.3 PRACTICAL ESTIMATION WITH SHORT ROLLOUTS

The Doob-$h$ guided rates in equation 12 depend on the harmonic function $h_t(x) = \mathbb{E}[G(X_1) \mid X_t = x]$, which is intractable to compute exactly in large edit spaces. pCoMole therefore approximates $h_t$ using short Monte Carlo rollouts under the base Edit Flow. Given $(t, x)$, we simulate $R$ independent continuations from time $t$ to 1 using the base rate field $\{u_s\}_{s \in [t,1]}$, producing terminal states $\{X_1^{(r)}(t, x)\}_{r=1}^R$, and define

$$\widehat{h}_t(x) \;:=\; \frac{1}{R} \sum_{r=1}^{R} G\Big(X_1^{(r)}(t, x)\Big). \tag{13}$$

When the continuations are exact simulations of the base CTMC, $\widehat{h}_t(x)$ is an unbiased estimator of $h_t(x)$, with feasibility incorporated through $G$.

Evaluating equation 13 for all admissible successors is prohibitive, so we adopt a two-stage approximation. At a decision point $(t, x_t)$, we first sample $C$ candidate next states $\{y_c\}_{c=1}^C$ by drawing $C$ edit events from the base rates $u_t(\cdot \mid x_t)$ and applying the corresponding edits. For each candidate $y_c$, we run $R$ rollouts from $(t, y_c)$ to time 1 and compute

$$\widehat{h}_t(y_c) \;:=\; \frac{1}{R} \sum_{r=1}^{R} G\Big(X_1^{(c,r)}\Big), \qquad c \in \{1, \dots, C\}. \tag{14}$$

Substituting $\widehat{h}_t$ into the Doob-$h$ reweighting yields an approximate guided selection rule over the candidate set,

$$\widetilde{u}_t(y_c \mid x_t) \;\propto\; u_t(y_c \mid x_t)\,\widehat{h}_t(y_c), \tag{15}$$

with normalization over $c \in \{1, \dots, C\}$. Here, $u_t(y_c \mid x_t)$ proposes locally plausible edits, while $\widehat{h}_t(y_c)$ favors candidates whose downstream continuations achieve higher terminal preference under $G$, naturally downweighting candidates unlikely to admit feasible completions.

Because short rollouts may miss feasible terminals under limited budgets, pCoMole maintains an incumbent feasible terminal sequence with the highest observed preference value across all rollouts. For efficiency, we first prune candidates using a cheap screening criterion and allocate rollouts only to candidates whose immediate scalarized score exceeds that of the current state. We then discard candidates whose best observed feasible terminal preference fails to improve upon the current sequence, focusing computation on candidates with evidence of downstream improvement. After the iterative editing phase, we run one additional terminal rollout from the best intermediate state and return the best feasible terminal sequence observed across the entire sampling process. The rollout approximation and pruning preserve the main guarantees of pCoMole (Appendix C).

## 4 EXPERIMENTS

Our experiments focus on molecule editing settings, where the goal is to improve multiple properties while preserving feasibility under a constrained edit process. In biomolecular design, reducing sequence length is a central practical objective to improve synthesizability, manufacturability, and delivery, whereas systematically increasing sequence length is typically misaligned with the design paradigm and offers limited utility. To demonstrate the efficacy of pCoMole, we evaluate on three shrinkage benchmarks: (i) peptidomimetic binder design, where long peptide binders are compressed

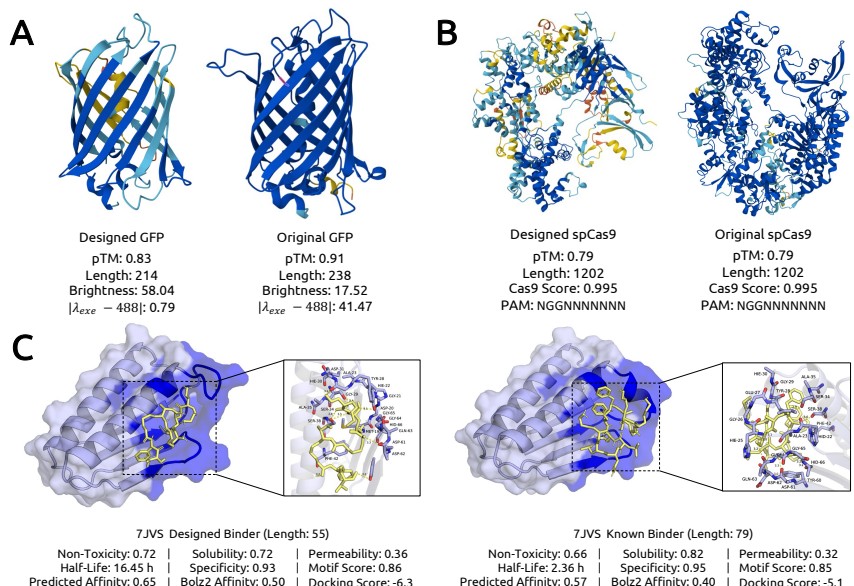

Figure 2: **pCoMole enables constraint-aware multi-objective editing across proteins and peptidomimetic binders. (A) GFP miniaturization.** A pCoMole-designed GFP preserves the canonical $\beta$-barrel while reducing length and improving optical objectives; excitation alignment to the 488 nm laser line is reported as $|\lambda_{\text{exc}} - 488|$. **(B) Cas9 shrinkage.** A pCoMole-designed spCas9 satisfies the PAM constraint and achieves a high Cas9-likeness score while maintaining a plausible structure (pTM). **(C) Short peptidomimetic binder design.** pCoMole shrinks the 7JVS binder while improving predicted properties; the binder is shown in yellow, the target in light blue, and the dark-blue surface highlights the target motif region(s). Insets zoom into the binding interface.

into shorter peptidomimetics while improving multiple drug-related properties; (ii) GFP shrinkage, which designs shorter fluorescent proteins without sacrificing fluorescence-related objectives; and (iii) Cas9 shrinkage, which aims to reduce protein length while preserving PAM specificity.

## 4.1 EDIT FLOWS ENABLE HIGH-QUALITY BIOLOGICAL SEQUENCE EDITING

To support sequence editing, we first trained five unconditional Edit Flow models: (i) a UniRef short-sequence Edit Flow, $\text{UniRef}_S$, trained on around 20k proteins no longer than 350 residues; (ii) a GFP Edit Flow obtained by fine-tuning $\text{UniRef}_S$ on around 800 known GFP sequences; (iii) a UniRef long-sequence Edit Flow, $\text{UniRef}_L$, trained on around 40k proteins within 800 residues to 1600 residues in length; (iv) a Cas9 Edit Flow trained from scratch on around 20k Cas9-family sequences; (v) a peptidomimetic Edit Flow trained from scratch on 28k peptidomimetics represented as SELFIES strings. These editors define variable-length CTMC dynamics through insertion, deletion, and substitution operations, serving as the fixed proposal processes for pCoMole to steer toward feasibility and user preferences.

Our Edit Flow models achieve low train-test gaps at their respective length scales, obtaining a lower loss when trained on domain-specific data (Table 1). To assess unconditional generation quality, we sample 1,000 sequences from each Edit Flow model and report validity and diversity (Appendix E.3). We further compare structural confidence by plotting the distribution of mean pLDDT scores for 1,000 sequences generated by the $\text{UniRef}_S$ Edit Flow against the corresponding 1,000 $\text{UniRef}_S$ test sequences used as inputs (Figure S1). The generated sequences closely match the pLDDT distribution of the originals, indicating that sampling preserves overall predicted fold quality.

## 4.2 PCOMOLE SHRINKS GFP WHILE OPTIMIZING OPTICAL FUNCTIONS UNDER EXPERIMENTAL CONSTRAINTS

We evaluate pCoMole on shrinking GFP while preserving or improving optical function. Shorter GFPs can reduce expression burden and improve experimental deployability, but fluorescence requires

Table 1: **Edit Flow training and evaluation metrics across sequence domains.** Validity rate is not applicable for UniRef$_S$/UniRef$_L$.

| Edit Flow | Train Loss | Test Loss | Diversity | Validity Rate |
|---|---|---|---|---|
| UniRef$_S$ | 594.39 | 557.44 | 0.68 | — |
| UniRef$_L$ | 3620.36 | 3610.61 | 0.41 | — |
| GFP | 188.87 | 501.58 | 0.54 | 0.95 |
| Cas9 | 2430.42 | 2527.12 | 0.25 | 0.90 |
| Peptidomimetic | 36.26 | 38.40 | 0.80 | 0.70 |

Table 2: **GFP design comparison between pCoMole, RayGun, and SCISOR under a fixed length constraint.** For each method, we generate 100 candidates and report average metrics. All methods enforce a fixed terminal length of 213 residues and achieve a near 100% validity rate.

| Method | $\|\lambda_{\mathrm{exc}} - 488\|$ ($\downarrow$) | Brightness |
|---|---|---|
| RayGun | 22.3594 | 9.4207 |
| SCISOR | 20.1279 | 19.0333 |
| pCoMole (UniRef$_S$ Edit Flow) | **5.8213** | 43.5116 |
| pCoMole (GFP Edit Flow) | 8.4372 | **49.0121** |

maintaining the characteristic $\beta$-barrel fold, making shrinkage challenging for sequence-based methods. We guide pCoMole to (i) increase predicted brightness, (ii) align the predicted excitation peak to the common 488 nm laser line, and (iii) reduce length. We enforce three constraints: a GFP classifier constraint, an emission-range constraint requiring the predicted emission peak to remain in the green band, and a terminal length constraint (hard equality or soft bound). Brightness, excitation, and emission are predicted with FPredX (Tam & Zhang, 2022). All experiments start from the same GFP sequence (UniProt P42212; 238 residues) to enable consistent comparison and facilitate downstream testing.

We compare pCoMole against RayGun (Devkota et al., 2024) and SCISOR (Baron et al., 2025), two state-of-the-art protein shrinkage methods (Table 2). Since both require a fixed output length, we impose the same hard terminal length constraint for all methods and generate sequences of exactly 213 residues. Under this controlled setting, pCoMole yields substantially better optical metrics, producing candidates with excitation closer to 488 nm and markedly higher predicted brightness than either baseline. Notably, pCoMole remains competitive using a UniRef-pre-trained Edit Flow without GFP-specific fine-tuning, indicating that improvements arise from multi-objective, constraint-aware sampling rather than a specialized prior.

We next ablate objective guidance using a soft length constraint that limits shrinkage to at most 25 residues (minimum length 213), since excessive truncation can destabilize the $\beta$-barrel (Table S2). Joint optimization of brightness and excitation alignment provides the best trade-off. Removing either term causes the expected regression (lower brightness without brightness guidance, and excitation drift from 488 nm without excitation guidance). AlphaFold3 predictions for representative designs show preservation of the canonical $\beta$-barrel architecture alongside gains in targeted optical properties (Figure S2) (Abramson et al., 2024).

Because pCoMole incurs additional computation from rollouts and objective evaluation, we also compare methods under a fixed wall-clock budget rather than a fixed sample count. In the time pCoMole produces one design, SCISOR and RayGun generate 334 and 3655 candidates, respectively. We report the best-achieved scores among the top-$N$ candidates up to these pool sizes (Table S1). Despite large sampling advantages, shrinkage-only baselines plateau (SCISOR at brightness 40.3; RayGun at 16.7), whereas pCoMole consistently yields substantially brighter designs across five runs while maintaining strong excitation alignment. Thus, pCoMole provides better designs per unit time when objective- and constraint-aware editing matters, while fast shrinkers remain preferable for shrinkage-only goals. For the compute-matched comparison, we rerun all methods independently and enforce the same exact terminal length (213 residues) as in Table 2.

Table 3: **pCoMole shrinks common Cas9s while preserving PAM specificity.** 'Cas9 Likelihood' is a pre-trained classifier score. 'PAM Distribution CE' is the cross-entropy between input and generated PAM distributions. 'PAM Match Rate' is the fraction of samples with an exact PAM match. 'PAM matched' reports the preserved predicted PAM.

| Input Cas9 | Length Change | Cas9 Likelihood | PAM Distr. CE | PAM Match Rate | PAM matched |
|---|---|---|---|---|---|
| SpCas9 | $1368 \rightarrow 1211 \ (-157)$ | $1.00 \rightarrow 0.97$ | $0.97 \rightarrow 0.90$ | 1.00 | NGG |
| St1Cas9 | $1121 \rightarrow 1023 \ (-98)$ | $1.00 \rightarrow 0.98$ | $0.91 \rightarrow 0.89$ | 1.00 | NNANAA |
| St3Cas9 | $1388 \rightarrow 1301 \ (-87)$ | $1.00 \rightarrow 0.95$ | $0.94 \rightarrow 0.90$ | 1.00 | NGGNG |
| GeoCas9 | $1087 \rightarrow 1035 \ (-52)$ | $0.96 \rightarrow 0.95$ | $0.93 \rightarrow 0.89$ | 1.00 | NNNNCNAA |

### 4.3 PCOMOLE SHRINKS CAS9S WHILE MAINTAINING PAM SPECIFICITY

CRISPR-Cas9 enables precise genome editing, but the large size of Cas9 endonucleases limits delivery and packaging, especially for viral vectors (Edraki et al., 2019; Ran et al., 2015; Behr et al., 2021; Kabadi et al., 2024; Wang et al., 2020). Shrinkage is therefore desirable, but must preserve PAM specificity, which determines the set of genomic targets a Cas9 can recognize (Collias & Beisel, 2021; Chatterjee et al., 2018; 2020; Zhao et al., 2023). We task pCoMole with shrinking Cas9 sequences while preserving Cas9 validity and the input PAM specificity.

pCoMole optimizes three objectives: maximizing Cas9-likelihood, minimizing cross-entropy between the edited sequence's predicted PAM distribution and that of the input, and maximizing length reduction. We impose hard terminal constraints requiring exact PAM matching, detection of the HNH and RuvC catalytic domains, and a shorter terminal length. Cas9-likelihood is scored by a pre-trained classifier (valid Cas9 vs. UniRef, other Cas proteins, and corrupted variants), domain presence is checked via a simple HMM, and PAM prediction is performed with Protein2PAM (Nayfach et al., 2025).

We evaluate pCoMole on four well-characterized Cas9s, SpCas9, St1Cas9, St3Cas9, and GeoCas9, by sampling 50 shrunk variants per input (Table 3). Across all cases, pCoMole achieves substantial length reduction while maintaining high Cas9-likelihood. The PAM-matching terminal constraint ensures every generated sequence retains a predicted PAM identical to the original, indicating preserved target specificity.

We further benchmark against RayGun and SCISOR (Table S3) (Devkota et al., 2024; Baron et al., 2025). Since both baselines require a fixed output length, we shrink CjCas9 from 984 to 934 residues for all methods, generating 100 sequences each. pCoMole preserves the predicted PAM for all samples without loss of Cas9-likelihood. In contrast, only 35% of SCISOR samples retain the correct PAM and their Cas9-likelihood is degraded, while RayGun produces no sequences that preserve either PAM specificity or Cas9-likeness. This highlights pCoMole's ability to preserve functional specificity under aggressive shrinkage.

Finally, we ablate insertion operations (Table S4). Because the underlying Edit Flow is unconditional, insertions can counteract length reduction. Disabling insertions consistently yields greater shrinkage with negligible impact on Cas9-likelihood or PAM preservation, making insertion-free editing the preferred configuration for Cas9 shrinkage.

### 4.4 PCOMOLE DEVELOPS SHORT PEPTIDOMIMETIC BINDERS OPTIMIZING SEVEN DRUG-RELATED PROPERTIES

We apply pCoMole with a peptidomimetic Edit Flow to shrink known peptide binders into short peptidomimetic candidates. Peptidomimetics can mitigate key liabilities of linear peptides (stability, permeability, pharmacokinetics) while retaining binding-critical motifs. Starting from each binder, pCoMole performs edit-based shrinkage while optimizing seven objectives: non-toxicity, solubility, permeability, half-life, binding affinity, binding motif score, and motif specificity. The motif-based objectives encourage preservation of the original pocket by rewarding motif retention and penalizing off-target binding. We additionally impose a soft shrinkage constraint to prevent degenerate collapse and a hard peptidomimetic validity constraint. Property scores come from pre-trained predictors, and constraints are computed deterministically from SMILES using RDKit-based rules.

Table 4: **pCoMole designs short peptidomimetics for existing peptide drugs with improved properties.** For each target, we report the property scores of the original drug binders and the average scores over 100 designed peptidomimetics (rows labeled "pCoMole"). Length is the number of SMILES tokens after tokenization. All designed sequences satisfy the specified constraints. For GLP-1R, pCoMole shrinks Semaglutide, so the motif score and specificity are defined relative to its binding motif and shown only for Semaglutide and its pCoMole designs, with "—" for other GLP-1R drugs.

| Target | Ligand | Length | Non-Toxicity | Solubility | Permeability | Half-life (h) | Affinity | Motif Score | Specificity |
|--------|--------|--------|--------------|------------|--------------|---------------|----------|-------------|-------------|
| PTH1R | Teriparatide | 259 | 0.8890 | 0.6905 | 0.2793 | 2.8301 | 0.9148 | 0.2696 | 0.9834 |
|       | pCoMole | 217 | 0.8729 | 0.7268 | 0.2959 | 2.9488 | 0.9358 | 0.3593 | 0.9706 |
| p53 | p28 | 170 | 0.8533 | 0.6737 | 0.2627 | 2.2015 | 0.6369 | 0.5418 | 0.9873 |
|     | pCoMole | 133 | 0.8076 | 0.6309 | 0.3071 | 3.8517 | 0.6885 | 0.5014 | 0.9649 |
| GLP-1R | Semaglutide | 248 | 0.8872 | 0.7656 | 0.2583 | 1.8571 | 0.9467 | 0.4691 | 0.9827 |
|        | pCoMole | 186 | 0.8122 | 0.7240 | 0.2985 | 4.0277 | 0.9367 | 0.5212 | 0.9778 |
|        | Liraglutide | 237 | 0.7854 | 0.8307 | 0.2609 | 2.0808 | 0.8440 | — | — |
|        | Exenatide | 256 | 0.8319 | 0.7547 | 0.2319 | 2.0663 | 0.8949 | — | — |
|        | Lixisenatide | 286 | 0.7204 | 0.8392 | 0.2215 | 2.7535 | 0.8280 | — | — |
|        | Tirzepatide | 273 | 0.9491 | 0.8303 | 0.2573 | 1.7149 | 0.9798 | — | — |

We evaluate pCoMole on 12 PDB targets with known peptide binders (Table S5), generating 100 candidates per target. All designs satisfy both terminal constraints and achieve substantial length reduction. Beyond compression, candidates typically improve permeability, predicted half-life, and predicted affinity relative to the original peptides, while largely preserving (and often improving) motif and specificity scores, suggesting shrinkage does not disrupt pocket engagement. As expected, improving permeability, half-life, or affinity can entail modest decreases in predicted non-toxicity or solubility for some targets, reflecting realistic multi-objective trade-offs.

To assess binding plausibility, we perform structure-based analyses on representative designs (Figure 2C, Figures S3-S6). For each target, we select a design with strong Boltz-2 predicted affinity and dock it with AutoDock VINA (Passaro et al., 2025; Trott & Olson, 2010). Docked poses consistently place designs in the same pocket as the known peptides, adjacent to annotated motif regions, and engaging highly overlapping residue sets, supporting that pCoMole preserves pocket-level recognition under an independent structure-based evaluation.

Finally, we apply pCoMole to clinically relevant peptide drugs on three therapeutic targets (Table 4): teriparatide for PTH1R (Neer et al., 2001), p28 for p53 (Warso et al., 2013; Yamada et al., 2013), and semaglutide for GLP-1R (with comparisons to other GLP-1R drugs) (Marso et al., 2016; Wilding et al., 2021). Across targets, pCoMole yields substantially shorter, constraint-satisfying peptidomimetics with improved permeability and half-life while maintaining strong predicted affinity, again exhibiting expected trade-offs in non-toxicity or solubility. Notably, GLP-1R designs derived from semaglutide are competitive with, and often outperform, established GLP-1R drugs across these predicted property profiles.

A detailed discussion of ablation studies is provided in Appendix Section D.

## 5 DISCUSSION

We present **pCoMole**, a framework for constraint-aware, multi-objective *sequence editing* that steers a pre-trained Edit Flow toward feasible, Pareto-aligned outcomes under hard terminal constraints. By operating on variable-length edit trajectories rather than generating sequences *de novo*, pCoMole enables systematic shrinkage while preserving task-critical properties. We validate this capability across protein and small-molecule settings, including GFP shrinkage with *in silico* evaluation of fluorescence-related objectives. Complementary *in silico* results on diverse Cas9 orthologs and peptidomimetic binder compression further show that pCoMole can maintain stringent specifications such as PAM preference or binding motifs while navigating realistic multi-objective trade-offs, with wet-lab validation representing an important next step. Despite higher computational cost from rollout-based guidance, pCoMole provides a principled bridge between discrete generative modeling and biomolecular design.

**Reproducibility.** Our code is available for review at here.

## MEANINGFULNESS STATEMENT

pCoMole helps learn meaningful representations of life by framing biomolecular design as constraint-aware, multi-objective editing of real biological sequences rather than unconstrained de novo generation. By steering Edit Flows with hard feasibility constraints and Pareto trade-off control, the model must capture which sequence changes preserve fold, function, and specificity (e.g. GFP optics, Cas9 PAM recognition) while allowing large structural edits such as deletions and insertions. This yields representations that are sensitive to biological invariances and failure modes, and that connect local edits to system-level phenotypes. Such edit-grounded representations can support more interpretable, testable, and transferable models for engineering proteins and therapeutics.

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

# Appendix

## A  RELATED WORK

**Molecule Editing Frameworks.**   Protein miniaturization methods such as RayGun (Devkota et al., 2024) and SCISOR (Baron et al., 2025) focus on producing shorter proteins, but are not designed for multi-objective editing that simultaneously optimizes developability and functional criteria. We benchmark both against pCoMole in Sections 4.2 and 4.3. In contrast, many small-molecule editing and optimization methods, including MolDQN (Zhou et al., 2019) and GCPN (You et al., 2018), are formulated as online reinforcement learning, relying on repeated reward evaluation during training or search. pCoMole instead operates in an offline setting with costly per-sequence evaluation, making direct comparison to online methods inappropriate.

**Constraint-based Generation.**   Constraint-aware generative methods typically enforce feasibility through projection, constrained dynamics, or constraint-driven fine-tuning. Representative diffusion and flow-matching approaches include CDD (Cardei et al., 2025), DiffOPT (Kong et al., 2024), and ALF$^2$ (Gutjahr et al., 2025). However, these methods are not formulated as sequence-editing processes with explicit insertion and deletion for variable-length control, and they generally optimize a single scalar objective with constraint penalties rather than providing explicit Pareto-style multi-objective control. Many recent constrained protein design methods further rely on structure-guided constraints, differing from pCoMole's sequence-only editing setting.

**Multi-Objective Optimization.**   Multi-objective guided generation is commonly studied in sequential design settings, where candidates are iteratively proposed and refined using new evaluations (Gruver et al., 2023; Jain et al., 2023; Stanton et al., 2022; Ahmadianshalchi et al., 2024). Bayesian optimization methods fit surrogate models and select candidates using acquisition functions (Yu et al., 2020; Shahriari et al., 2015), with multi-objective extensions based on expected hypervolume improvement (Emmerich & Klinkenberg, 2008), information-theoretic criteria (Belakaria et al., 2021), or scalarization strategies (Knowles, 2006; Zhang & Li, 2007; Paria et al., 2020). While effective for black-box trade-off exploration, these approaches typically treat feasibility through soft penalties or heuristics and do not model feasibility-gated sampling under hard biochemical or manufacturability constraints.

**Constrained Multi-Objective Molecular Optimization.**   Recent frameworks such as CMOMO (Xia et al., 2025) and MCEMOL (Lin et al., 2026) study constrained multi-objective molecular optimization, but target different pipelines than ours. CMOMO performs iterative evolutionary search over RDKit-based fitness functions, which does not transfer to sequence editing under biochemical and manufacturability constraints. We do not benchmark against MCEMOL due to the lack of a public implementation. Other constrained multi-objective methods based on search or reinforcement learning, including MolSearch (Sun et al., 2022), ParetoDrug (Yang et al., 2024), and REINVENT 4 (Loeffler et al., 2024), likewise assume repeated online evaluations, whereas pCoMole performs offline, edit-based sampling from a fixed pre-trained editor.

## B  LOG-DOMAIN ROLLOUT ESTIMATION AND GUIDANCE

Our guided editing procedure relies on a terminal preference function of the form $G(x) = \exp(\ell(x))$, where $\ell(x)$ is a scalar score derived from multi-objective utility and feasibility constraints. In practice, objective scores may be unbounded or have large magnitude, and direct computation of $\exp(\ell(x))$ can overflow. We therefore implement value estimation and guided selection entirely in the log domain, which removes the need to normalize objectives to a fixed range and yields numerically stable computations.

**Log-preference and feasibility.**   Let $U(x)$ denote the scalar utility used for guidance, such as the augmented Tchebycheff utility defined in the main text. We define the log-preference as

$$\ell(x) := \beta U(x) + \log \mathbb{I}[x \in \mathcal{F}], \tag{16}$$

where $\beta > 0$ is an inverse-temperature and $\mathbb{I}[x \in \mathcal{F}]$ is the feasibility indicator. We implement the indicator in log space by setting $\log \mathbb{I}[x \in \mathcal{F}] = 0$ for feasible $x$ and $-\infty$ otherwise. This convention ensures that infeasible terminal states contribute zero weight in all subsequent computations.

**Log-domain value function and Monte Carlo estimator.** Doob-$h$ guidance depends on the space-time harmonic function $h_t(x) = \mathbb{E}[G(X_1) \mid X_t = x]$. In log space, we define

$$\log h_t(x) \;=\; \log \mathbb{E}[\exp(\ell(X_1)) \mid X_t = x]. \tag{17}$$

Given $R$ independent rollouts from $(t, x)$ under the base Edit Flow, producing terminal states $\{X_1^{(r)}\}_{r=1}^R$, the standard Monte Carlo estimator of $h_t(x)$ is $\widehat{h}_t(x) = \frac{1}{R} \sum_{r=1}^R \exp(\ell(X_1^{(r)}))$. We compute its logarithm stably using the log-mean-exp operator

$$\log \widehat{h}_t(x) \;=\; \log \sum_{r=1}^R \exp(\ell(X_1^{(r)})) \;-\; \log R, \tag{18}$$

where the summation is implemented via logsumexp. Although $\log \widehat{h}_t(x)$ is generally a biased estimator of $\log h_t(x)$ due to Jensen's inequality, it is consistent as $R \to \infty$ and remains well-defined even when $\ell(X_1^{(r)})$ takes large values. When all rollouts terminate in infeasible states, equation 18 evaluates to $-\infty$, which correctly assigns zero continuation value under the feasibility-weighted preference.

**Guided selection from candidate edits in log space.** At time $t$ and state $x_t$, we restrict attention to a candidate set $\{y_j\}_{j=1}^M$ generated by sampling admissible edits from the base rate field $u_t(\cdot \mid x_t)$. Under the Doob-$h$ transform, the guided rate for a transition $x_t \to y$ is proportional to $u_t(y \mid x_t) h_t(y)$. Replacing $h_t$ by the rollout estimator and working in log space yields the candidate score

$$s_j \;:=\; \log u_t(y_j \mid x_t) \;+\; \log \widehat{h}_t(y_j). \tag{19}$$

We sample the next state from the categorical distribution obtained by normalizing these scores,

$$\mathbb{P}(Y = y_j \mid x_t) \;=\; \frac{\exp(s_j)}{\sum_{m=1}^M \exp(s_m)} \;=\; \exp\Big(s_j - \mathrm{logsumexp}_{m \in \{1, \ldots, M\}} \, s_m\Big), \tag{20}$$

which is computed stably by subtracting $\max_m s_m$ before exponentiation. This computation depends only on differences of log scores and is therefore invariant to adding a constant offset to $\ell(x)$ or to all scores $\{s_j\}$, which is useful when utilities are defined on arbitrary scales.

**Stability considerations.** The log-domain implementation prevents overflow in two places where it commonly occurs. First, equation 18 avoids forming $\exp(\ell)$ explicitly by using logsumexp. Second, equation 20 avoids overflow and underflow when converting scores into probabilities by applying the standard stabilized softmax. In addition, infeasible rollouts propagate cleanly as $-\infty$ scores and therefore receive zero probability without requiring special-case branching. These properties allow pCoMole to operate with unnormalized objective scores and large-magnitude utilities while preserving the same guided selection rule as the corresponding computations in the original domain.

## C   THEORETICAL PROOFS

This section states guarantees for the idealized pCoMole algorithm that uses exact Doob-$h$ guidance and exact value functions, and then describes what properties are preserved under the rollout-based approximations and practical safeguards used in pCoMole. We follow the notations in Section 3.

**Proposition C.1** (Doob-$h$ transform yields preference-tilted terminals). *Assume $G(X_1)$ is integrable under the base Edit Flow and $h_t(x) > 0$ for all $(t, x)$ reachable from $x_0$, where $h_t(x) := \mathbb{E}[G(X_1) \mid X_t = x]$. The CTMC with rates equation 12 has terminal distribution $q_G(\cdot \mid x_0)$ defined in equation 10.*

*Proof.* Let $(X_t)_{t \in [0,1]}$ be the base time-inhomogeneous CTMC on $\mathcal{X}$ with off-diagonal rates $u_t(y \mid x)$ and initial condition $X_0 = x_0$. Let $(\Omega, \mathcal{F}, \mathbb{P})$ denote its probability space and let $\mathcal{F}_t = \sigma(X_s : 0 \leq s \leq t)$ be the natural filtration. For $t \in [0, 1]$ and $x \in \mathcal{X}$ define

$$h_t(x) := \mathbb{E}_{\mathbb{P}}[G(X_1) \mid X_t = x],$$

which is well-defined by integrability of $G(X_1)$ and strictly positive on the reachable set by assumption. Define the likelihood-ratio process

$$L_t := \frac{h_t(X_t)}{h_0(x_0)}, \qquad t \in [0,1]. \tag{21}$$

By the Markov property of $X$, for any $0 \le s \le t \le 1$,

$$\mathbb{E}_{\mathbb{P}}[h_t(X_t) \mid \mathcal{F}_s] = \mathbb{E}_{\mathbb{P}}[\mathbb{E}_{\mathbb{P}}[G(X_1) \mid X_t] \mid \mathcal{F}_s] = \mathbb{E}_{\mathbb{P}}[G(X_1) \mid \mathcal{F}_s] = \mathbb{E}_{\mathbb{P}}[G(X_1) \mid X_s] = h_s(X_s), \tag{22}$$

so $(h_t(X_t))_{t \in [0,1]}$ is a $\mathbb{P}$-martingale and therefore $(L_t)_{t \in [0,1]}$ is a nonnegative $\mathbb{P}$-martingale with $\mathbb{E}_{\mathbb{P}}[L_t] = 1$. We use $L_1$ to define a new measure $\mathbb{Q}$ on $(\Omega, \mathcal{F}_1)$ by

$$\frac{d\mathbb{Q}}{d\mathbb{P}} := L_1 = \frac{h_1(X_1)}{h_0(x_0)} = \frac{G(X_1)}{h_0(x_0)}, \tag{23}$$

where we used $h_1(x) = \mathbb{E}_{\mathbb{P}}[G(X_1) \mid X_1 = x] = G(x)$. For any measurable set $A \subseteq \mathcal{X}$,

$$\mathbb{Q}(X_1 \in A) = \mathbb{E}_{\mathbb{P}}\left[\mathbb{I}[X_1 \in A] \frac{G(X_1)}{h_0(x_0)}\right] = \frac{1}{h_0(x_0)} \sum_{x_1 \in A} \mathbb{P}(X_1 = x_1) \, G(x_1), \tag{24}$$

which shows that the terminal marginal under $\mathbb{Q}$ is exactly the tilted distribution in equation 10.

It remains to identify the dynamics of $(X_t)$ under $\mathbb{Q}$. Fix $t \in [0,1)$ and a reachable state $x$. For any $y \ne x$, the base CTMC satisfies

$$\mathbb{P}(X_{t+\Delta} = y \mid X_t = x) = u_t(y \mid x) \, \Delta + o(\Delta), \qquad \Delta \downarrow 0. \tag{25}$$

Using Bayes' rule under the change of measure equation 23 and the Markov property,

$$\mathbb{Q}(X_{t+\Delta} = y \mid X_t = x) = \frac{\mathbb{E}_{\mathbb{P}}[\mathbb{I}[X_{t+\Delta} = y] L_1 \mid X_t = x]}{\mathbb{E}_{\mathbb{P}}[L_1 \mid X_t = x]} = \frac{\mathbb{E}_{\mathbb{P}}\left[\mathbb{I}[X_{t+\Delta} = y] \frac{G(X_1)}{h_0(x_0)} \mid X_t = x\right]}{\mathbb{E}_{\mathbb{P}}\left[\frac{G(X_1)}{h_0(x_0)} \mid X_t = x\right]}$$

$$= \frac{\mathbb{P}(X_{t+\Delta} = y \mid X_t = x) \, \mathbb{E}_{\mathbb{P}}[G(X_1) \mid X_{t+\Delta} = y]}{\mathbb{E}_{\mathbb{P}}[G(X_1) \mid X_t = x]} \tag{26}$$

$$= \frac{\mathbb{P}(X_{t+\Delta} = y \mid X_t = x) \, h_{t+\Delta}(y)}{h_t(x)}. \tag{27}$$

Since $h_{t+\Delta}(y) = h_t(y) + o(1)$ as $\Delta \downarrow 0$ for fixed $y$, combining equation 25 and equation 27 yields

$$\mathbb{Q}(X_{t+\Delta} = y \mid X_t = x) = u_t(y \mid x) \frac{h_t(y)}{h_t(x)} \Delta + o(\Delta).$$

Therefore, under $\mathbb{Q}$ the process is a CTMC with off-diagonal rates $u_t^G(y \mid x) = u_t(y \mid x) \, h_t(y)/h_t(x)$, which are precisely the Doob-$h$ guided rates in equation 12. Together with equation 24, this proves that the CTMC driven by equation 12 has terminal distribution $q_G(\cdot \mid x_0)$. $\quad\square$

**Proposition C.2** (ATC scalarization: Pareto optimality and coverage). *Let $\mathcal{F} \subseteq \mathcal{X}$ be the feasible set and let $U(x) = \min_{k \in \{1,\ldots,K\}} \omega_k(f_k(x) - r_k) + \rho \sum_{k=1}^K \omega_k(f_k(x) - r_k)$ with $\omega \in \mathbb{R}_{>0}^K$ and $\rho > 0$. If $x^\star \in \mathcal{F}$ maximizes $U(x)$ over $\mathcal{F}$, then $x^\star$ is Pareto-optimal in $\mathcal{F}$. Assume additionally that $f_k(x) > r_k$ for all $x \in \mathcal{F}$ and all $k$. Then for every Pareto-optimal $x^\star \in \mathcal{F}$, the weight choice*

$$\omega_k := \frac{1}{f_k(x^\star) - r_k}, \qquad k \in \{1, \ldots, K\}, \tag{28}$$

*makes $x^\star$ a maximizer of the Chebyshev term $T_\omega(x) := \min_k \omega_k(f_k(x) - r_k)$ over $\mathcal{F}$. Moreover, if $S_\omega(x) := \sum_{k=1}^K \omega_k(f_k(x) - r_k)$ is bounded above on $\mathcal{F}$ by $S_{\max}$ and the margin*

$$\delta := T_\omega(x^\star) - \sup_{x \in \mathcal{F} \setminus \{x^\star\}} T_\omega(x) \tag{29}$$

*is positive, then $x^\star$ also maximizes $U(x) = T_\omega(x) + \rho S_\omega(x)$ over $\mathcal{F}$ for all $\rho \in (0, \delta/(S_{\max} - S_\omega(x^\star)))$.*

*Proof.* Assume $x^\star \in \mathcal{F}$ maximizes $U$ for some $\omega \in \mathbb{R}_{>0}^K$ and $\rho > 0$. Suppose for contradiction that $x^\star$ is not Pareto-optimal. Then there exists $y \in \mathcal{F}$ such that $f_k(y) \geq f_k(x^\star)$ for all $k$ and $f_j(y) > f_j(x^\star)$ for at least one index $j$. Since $\omega_k > 0$, we have $\omega_k(f_k(y) - r_k) \geq \omega_k(f_k(x^\star) - r_k)$ for all $k$, which implies

$$\min_k \omega_k(f_k(y) - r_k) \geq \min_k \omega_k(f_k(x^\star) - r_k).$$

In addition, $\omega_j(f_j(y) - r_j) > \omega_j(f_j(x^\star) - r_j)$ and all other summands are nondecreasing, so

$$\sum_{k=1}^K \omega_k(f_k(y) - r_k) > \sum_{k=1}^K \omega_k(f_k(x^\star) - r_k).$$

Multiplying the strict inequality by $\rho > 0$ and adding the inequality for the minimum term yields $U(y) > U(x^\star)$, contradicting maximality of $x^\star$. Therefore $x^\star$ is Pareto-optimal.

For the coverage statement, fix a Pareto-optimal $x^\star \in \mathcal{F}$ and assume $f_k(x) > r_k$ for all $x \in \mathcal{F}$ and all $k$, so the weights in equation 28 are well-defined and positive. For this choice, $\omega_k(f_k(x^\star) - r_k) = 1$ for every $k$, hence $T_\omega(x^\star) = 1$. Consider any $x \in \mathcal{F}$ with $x \neq x^\star$. Since $x^\star$ is Pareto-optimal, $x$ cannot Pareto-dominate $x^\star$, so there exists at least one index $k$ with $f_k(x) < f_k(x^\star)$. For that index,

$$\omega_k(f_k(x) - r_k) < \omega_k(f_k(x^\star) - r_k) = 1,$$

which implies $T_\omega(x) = \min_j \omega_j(f_j(x) - r_j) < 1 = T_\omega(x^\star)$. Therefore $x^\star$ maximizes $T_\omega$ over $\mathcal{F}$.

Finally, assume $S_\omega$ is bounded above on $\mathcal{F}$ by $S_{\max}$ and the margin $\delta$ in equation 29 is positive. For any $x \in \mathcal{F} \setminus \{x^\star\}$, we have $T_\omega(x) \leq T_\omega(x^\star) - \delta$ and $S_\omega(x) \leq S_{\max}$, so

$$U(x) = T_\omega(x) + \rho S_\omega(x) \leq T_\omega(x^\star) - \delta + \rho S_{\max}.$$

On the other hand, $U(x^\star) = T_\omega(x^\star) + \rho S_\omega(x^\star)$. If $\rho < \delta/(S_{\max} - S_\omega(x^\star))$, then $T_\omega(x^\star) - \delta + \rho S_{\max} < T_\omega(x^\star) + \rho S_\omega(x^\star) = U(x^\star)$, which shows $U(x) < U(x^\star)$ for all $x \neq x^\star$ and therefore $x^\star$ maximizes $U$ over $\mathcal{F}$. $\qquad\square$

**Consistency of rollout-based guidance.** The practical algorithm replaces $h_t(x) = \mathbb{E}[G(X_1) \mid X_t = x]$ by the Monte Carlo estimator $\widehat{h}_t(x)$ and restricts attention to a finite candidate set $\mathcal{C}(x) \subset \mathcal{E}(x)$. The following statement controls the one-step error induced by multiplicative approximation of $h_t$ on $\mathcal{C}(x)$.

**Proposition C.3** (One-step kernel stability). *Fix $(t, x)$ and a candidate set $\mathcal{C}(x) \subset \mathcal{E}(x)$. Let $P(\cdot \mid x)$ be the exact Doob-$h$ transition distribution restricted to $\mathcal{C}(x)$, defined by*

$$P(y \mid x) = \frac{u_t(y \mid x)\, h_t(y)}{\sum_{y' \in \mathcal{C}(x)} u_t(y' \mid x)\, h_t(y')}, \qquad y \in \mathcal{C}(x).$$

*Let $\widehat{P}(\cdot \mid x)$ be the rollout-based transition distribution defined by*

$$\widehat{P}(y \mid x) = \frac{u_t(y \mid x)\, \widehat{h}_t(y)}{\sum_{y' \in \mathcal{C}(x)} u_t(y' \mid x)\, \widehat{h}_t(y')}, \qquad y \in \mathcal{C}(x).$$

*If there exists $\varepsilon \in [0, 1)$ such that*

$$(1 - \varepsilon)\, h_t(y) \leq \widehat{h}_t(y) \leq (1 + \varepsilon)\, h_t(y) \qquad \text{for all } y \in \mathcal{C}(x), \tag{30}$$

*then for all $y \in \mathcal{C}(x)$,*

$$\frac{1 - \varepsilon}{1 + \varepsilon} \leq \frac{\widehat{P}(y \mid x)}{P(y \mid x)} \leq \frac{1 + \varepsilon}{1 - \varepsilon}. \tag{31}$$

*Proof.* Fix $(t, x)$ and abbreviate $\mathcal{C} = \mathcal{C}(x)$. For each $y \in \mathcal{C}$, define the unnormalized weights

$$a_y := u_t(y \mid x)\, h_t(y), \qquad \widehat{a}_y := u_t(y \mid x)\, \widehat{h}_t(y),$$

and the corresponding normalizing constants

$$Z := \sum_{y' \in \mathcal{C}} a_{y'}, \qquad \widehat{Z} := \sum_{y' \in \mathcal{C}} \widehat{a}_{y'}.$$

By definition, $P(y \mid x) = a_y/Z$ and $\widehat{P}(y \mid x) = \widehat{a}_y/\widehat{Z}$.

Assumption equation 30 implies $(1 - \varepsilon)a_y \leq \widehat{a}_y \leq (1 + \varepsilon)a_y$ for all $y \in \mathcal{C}$. Summing these inequalities over $y \in \mathcal{C}$ yields

$$(1 - \varepsilon)Z \leq \widehat{Z} \leq (1 + \varepsilon)Z. \tag{32}$$

Therefore, for any $y \in \mathcal{C}$,

$$\frac{\widehat{P}(y \mid x)}{P(y \mid x)} = \frac{\widehat{a}_y}{a_y} \cdot \frac{Z}{\widehat{Z}}.$$

The first factor satisfies $(1 - \varepsilon) \leq \widehat{a}_y/a_y \leq (1 + \varepsilon)$, and equation 32 implies $\frac{1}{1+\varepsilon} \leq Z/\widehat{Z} \leq \frac{1}{1-\varepsilon}$. Multiplying the corresponding lower and upper bounds gives equation 31. □

The multiplicative condition in equation 30 follows from finite-rollout estimation under mild moment assumptions. The next proposition provides a sufficient condition based on a variance bound.

**Proposition C.4** (Finite-rollout multiplicative accuracy). *Fix $(t, x)$ and a finite candidate set $\mathcal{C}(x)$. For each $y \in \mathcal{C}(x)$, let $Z^{(y)} := G(X_1)$ under the conditional law $(X_t = y)$, and assume $\mathbb{E}[Z^{(y)}] = h_t(y) > 0$ and $\mathrm{Var}(Z^{(y)}) = \sigma_t^2(y) < \infty$. Let $\widehat{h}_t(y) = \frac{1}{R}\sum_{r=1}^R Z_r^{(y)}$ be the Monte Carlo estimator formed from $R$ independent rollouts from $(t, y)$. Then for any $\varepsilon \in (0, 1)$,*

$$\mathbb{P}\Big( |\widehat{h}_t(y) - h_t(y)| \geq \varepsilon\, h_t(y) \Big) \leq \frac{\sigma_t^2(y)}{R\, \varepsilon^2\, h_t(y)^2}. \tag{33}$$

*Moreover, for any $\delta \in (0, 1)$,*

$$\mathbb{P}\Big( (1 - \varepsilon)h_t(y) \leq \widehat{h}_t(y) \leq (1 + \varepsilon)h_t(y) \text{ for all } y \in \mathcal{C}(x) \Big) \geq 1 - \delta \tag{34}$$

*whenever*

$$R \geq \frac{|\mathcal{C}(x)|}{\delta\, \varepsilon^2} \max_{y \in \mathcal{C}(x)} \frac{\sigma_t^2(y)}{h_t(y)^2}. \tag{35}$$

*Proof.* Fix $y \in \mathcal{C}(x)$. By construction, $\widehat{h}_t(y)$ is the average of $R$ independent copies of $Z^{(y)}$, hence $\mathbb{E}[\widehat{h}_t(y)] = h_t(y)$ and $\mathrm{Var}(\widehat{h}_t(y)) = \sigma_t^2(y)/R$. Chebyshev's inequality gives

$$\mathbb{P}\Big( |\widehat{h}_t(y) - h_t(y)| \geq \varepsilon\, h_t(y) \Big) \leq \frac{\mathrm{Var}(\widehat{h}_t(y))}{\varepsilon^2 h_t(y)^2} = \frac{\sigma_t^2(y)}{R\, \varepsilon^2\, h_t(y)^2},$$

which proves equation 33.

For the uniform statement, define the event $A_y = \{|\widehat{h}_t(y) - h_t(y)| \geq \varepsilon h_t(y)\}$. A union bound implies

$$\mathbb{P}\left( \bigcup_{y \in \mathcal{C}(x)} A_y \right) \leq \sum_{y \in \mathcal{C}(x)} \mathbb{P}(A_y) \leq \sum_{y \in \mathcal{C}(x)} \frac{\sigma_t^2(y)}{R\, \varepsilon^2\, h_t(y)^2} \leq \frac{|\mathcal{C}(x)|}{R\, \varepsilon^2} \max_{y \in \mathcal{C}(x)} \frac{\sigma_t^2(y)}{h_t(y)^2}.$$

Under equation 35, the right-hand side is at most $\delta$, which yields equation 34 because $\bigcap_{y \in \mathcal{C}(x)} A_y^c$ is exactly the event that equation 30 holds for all $y \in \mathcal{C}(x)$. □

Proposition C.3 is a one-step statement. The following proposition shows how one-step kernel stability propagates to the multi-step editing procedure when the algorithm is viewed as a discrete sequence of decision steps.

**Proposition C.5** (Propagation of one-step stability to multi-step trajectories). *Consider a discrete-time process $\{X^{(s)}\}_{s=0}^S$ on $\mathcal{X}$ with initial state $X^{(0)} = x^{(0)}$. At each step $s \in \{0, \ldots, S - 1\}$ and state $x$, a candidate set $\mathcal{C}_s(x) \subseteq \mathcal{X}$ is formed and the next state is sampled from a transition kernel supported on $\mathcal{C}_s(x)$. Let $P_s(\cdot \mid x)$ denote the ideal Doob-h transition distribution restricted to $\mathcal{C}_s(x)$ and let $\widehat{P}_s(\cdot \mid x)$ denote the rollout-based transition distribution restricted to the same candidate*

*set. Assume there exists $\varepsilon \in [0, 1)$ such that for every step $s$, every state $x$ reachable under the ideal process, and every $y \in \mathcal{C}_s(x)$ with $P_s(y \mid x) > 0$,*

$$\frac{1-\varepsilon}{1+\varepsilon} \leq \frac{\widehat{P}_s(y \mid x)}{P_s(y \mid x)} \leq \frac{1+\varepsilon}{1-\varepsilon}. \tag{36}$$

*Then for any trajectory $\tau = (x^{(0)}, x^{(1)}, \ldots, x^{(S)})$ such that $\prod_{s=0}^{S-1} P_s(x^{(s+1)} \mid x^{(s)}) > 0$,*

$$\left(\frac{1-\varepsilon}{1+\varepsilon}\right)^S \leq \frac{\widehat{\mathbb{P}}(\tau)}{\mathbb{P}(\tau)} \leq \left(\frac{1+\varepsilon}{1-\varepsilon}\right)^S, \tag{37}$$

*where*

$$\mathbb{P}(\tau) := \prod_{s=0}^{S-1} P_s(x^{(s+1)} \mid x^{(s)}), \qquad \widehat{\mathbb{P}}(\tau) := \prod_{s=0}^{S-1} \widehat{P}_s(x^{(s+1)} \mid x^{(s)}).$$

*Moreover, for any terminal state $x^{(S)}$ with $\mathbb{P}(X^{(S)} = x^{(S)}) > 0$,*

$$\left(\frac{1-\varepsilon}{1+\varepsilon}\right)^S \leq \frac{\widehat{\mathbb{P}}(X^{(S)} = x^{(S)})}{\mathbb{P}(X^{(S)} = x^{(S)})} \leq \left(\frac{1+\varepsilon}{1-\varepsilon}\right)^S. \tag{38}$$

*Proof.* Fix a trajectory $\tau = (x^{(0)}, \ldots, x^{(S)})$ with $\mathbb{P}(\tau) > 0$. By definition of the path probabilities,

$$\frac{\widehat{\mathbb{P}}(\tau)}{\mathbb{P}(\tau)} = \prod_{s=0}^{S-1} \frac{\widehat{P}_s(x^{(s+1)} \mid x^{(s)})}{P_s(x^{(s+1)} \mid x^{(s)})}.$$

Since $\mathbb{P}(\tau) > 0$, each factor in the denominator is positive, and the assumption equation 36 applies to each step along the trajectory. Applying the lower and upper bounds in equation 36 termwise and multiplying over $s = 0, \ldots, S-1$ yields equation 37.

For the terminal bound, fix a terminal state $x^{(S)}$ with $\mathbb{P}(X^{(S)} = x^{(S)}) > 0$, and let $\mathcal{T}(x^{(S)})$ denote the set of length-$S$ trajectories ending at $x^{(S)}$ with positive $\mathbb{P}(\tau)$. Summing equation 37 over $\tau \in \mathcal{T}(x^{(S)})$ gives

$$\left(\frac{1-\varepsilon}{1+\varepsilon}\right)^S \sum_{\tau \in \mathcal{T}(x^{(S)})} \mathbb{P}(\tau) \leq \sum_{\tau \in \mathcal{T}(x^{(S)})} \widehat{\mathbb{P}}(\tau) \leq \left(\frac{1+\varepsilon}{1-\varepsilon}\right)^S \sum_{\tau \in \mathcal{T}(x^{(S)})} \mathbb{P}(\tau).$$

The left and right sums equal $\mathbb{P}(X^{(S)} = x^{(S)})$ and $\widehat{\mathbb{P}}(X^{(S)} = x^{(S)})$, respectively, by definition of the terminal marginal as the sum of path probabilities over trajectories ending at $x^{(S)}$. Dividing both sides by $\mathbb{P}(X^{(S)} = x^{(S)})$ yields equation 38. □

**Properties preserved by practical safeguards.** pCoMole maintains an incumbent feasible terminal sequence among all rollouts executed during sampling and returns the best feasible terminal encountered. Let $\mathcal{S}$ denote the multiset of terminal sequences produced by all rollouts performed during sampling, including the final terminal rollout. When $\mathcal{S} \cap \mathcal{F} \neq \varnothing$, the algorithm returns

$$x_{\text{out}} \in \arg\max_{x \in \mathcal{S} \cap \mathcal{F}} G(x), \tag{39}$$

with an arbitrary tie-breaking rule. When $\mathcal{S} \cap \mathcal{F} = \varnothing$, the algorithm applies a predefined fallback rule that does not affect the statement below.

**Proposition C.6** (Incumbent optimality among evaluated feasible terminals). *If $\mathcal{S} \cap \mathcal{F} \neq \varnothing$, then $x_{\text{out}} \in \mathcal{F}$ and $G(x_{\text{out}}) \geq G(x)$ for all $x \in \mathcal{S} \cap \mathcal{F}$. Moreover, any pruning rule that only removes candidate evaluations or rollouts, without altering the preference function $G$ or the incumbent update rule, preserves equation 39 with $\mathcal{S}$ interpreted as the multiset of terminal sequences actually evaluated after pruning.*

*Proof.* Assume $\mathcal{S} \cap \mathcal{F} \neq \varnothing$. By definition equation 39, $x_{\text{out}}$ is selected from $\mathcal{S} \cap \mathcal{F}$ and maximizes $G$ over this set, which implies $x_{\text{out}} \in \mathcal{F}$ and $G(x_{\text{out}}) \geq G(x)$ for all $x \in \mathcal{S} \cap \mathcal{F}$.

For the pruning statement, let $\mathcal{S}_{\text{full}}$ denote the multiset of terminal sequences that would be produced without pruning and let $\mathcal{S}_{\text{pruned}}$ denote the multiset of terminal sequences actually produced with pruning. A pruning rule that only removes evaluations can only decrease the set of executed rollouts, hence $\mathcal{S}_{\text{pruned}} \subseteq \mathcal{S}_{\text{full}}$ as multisets. The incumbent update rule is applied to the terminal sequences that are evaluated, so under pruning the algorithm returns an element of $\arg\max_{x \in \mathcal{S}_{\text{pruned}} \cap \mathcal{F}} G(x)$ whenever $\mathcal{S}_{\text{pruned}} \cap \mathcal{F} \neq \varnothing$. This is exactly equation 39 with $\mathcal{S}$ interpreted as the evaluated multiset after pruning, which completes the proof. □

**Approximate guarantees for rollout-based pCoMole with pruning.** The ideal Doob-$h$ construction yields the preference-tilted terminal law in Proposition C.1, and Pareto-optimality of ATC maximizers follows from Proposition C.2. The practical algorithm replaces $h_t$ by the rollout estimator $\widehat{h}_t$ and evaluates guidance on a sampled candidate set. Proposition C.4 shows that, with sufficiently many rollouts, $\widehat{h}_t$ satisfies the multiplicative accuracy condition equation 30 on a fixed candidate set with high probability, and Proposition C.3 then implies that the resulting one-step selection distribution is a controlled multiplicative perturbation of the ideal Doob-$h$ choice on that set. Proposition C.5 propagates this one-step stability across decision steps, implying that the terminal distribution induced by rollout-based guidance remains multiplicatively close to the ideal guided terminal distribution over any fixed editing horizon, and therefore approximately preserves the preference tilting that drives multi-objective improvement under a given weight vector.

Pruning and incumbent caching affect which rollouts are executed but do not alter the preference function $G$ used to evaluate terminals. Proposition C.6 guarantees that whenever any feasible terminal is generated by the executed rollouts, the returned output is feasible and maximizes $G$ among all feasible terminals that were evaluated. This safeguard preserves feasibility and best-observed preference within the evaluated set, while the approximation results above quantify how rollout-based guidance approaches the ideal Doob-$h$ behavior as rollout and candidate budgets increase. Coverage of distinct Pareto trade-offs is obtained by varying the weight vector across runs, and the same stability reasoning applies to each run independently, whereas pruning may reduce empirical coverage under a fixed computational budget by reducing the set of evaluated terminals.

# D   ABLATION STUDIES

**Constraint Ablation.** We ablate the terminal constraints used in peptidomimetic binder design to quantify their impact. (Table S6). The peptidomimetic hard constraint (PM) enforces chemical validity, while the soft length constraint (LEN) encourages shrinkage without reducing below half of the original size. Enforcing both constraints yields short, feasible designs with strong predicted properties and 100% satisfaction rates. Removing LEN leads to over-shrinking and degraded trade-offs, often sacrificing binding-related properties for extended half-life. Removing PM rarely produces invalid outputs due to the strong inductive bias of the underlying Edit Flow, but explicit enforcement provides an additional safety layer against off-manifold samples. Removing both constraints results in degenerate behavior, including extreme length collapse and loss of affinity and motif consistency, underscoring the necessity of terminal constraints for controlled shrinkage.

**Hyperparameter Ablation.** We evaluate sensitivity to three sampling hyperparameters: number of sampling steps, rollouts per candidate, and candidates per step (Tables S7, S8, S9). Increasing any budget improves shrinkage quality and predicted properties, particularly affinity and motif preservation, but with clear diminishing returns and increased runtime. More steps enable progressive refinement, additional rollouts reduce myopic decisions at higher cost, and more candidates improve exploration but saturate beyond moderate values. These trends motivate intermediate settings for all experiments (Appendix G). pCoMole is relatively robust to these choices because Doob-$h$ guidance concentrates probability mass on high-utility trajectories, so increased budgets often refine already strong preferences.

**Pareto Coverage Under Different Scalarizations.** We assess the effect of scalarization and weight choice on Pareto coverage for PDB 4O56, optimizing solubility, permeability, and binding affinity (Table S10). Across representative weight vectors, varying weights produce the expected trade-offs, confirming controllable navigation of the Pareto front. The augmented Tchebycheff (ATC) utility achieves the highest empirical coverage against a pooled reference front, while standard Tchebycheff

yields moderate coverage and linear weighted sums perform worst, consistent with their tendency to under-explore non-convex regions. These results support ATC as the default scalarization in pCoMole.

# E    EDIT FLOW DETAILS

## E.1    DATASET CURATION

**UniRef Dataset.**    We trained our general protein models on two datasets of UniRef sequences with differing lengths. To curate the dataset for our UniRef$_S$ Edit Flow, we sampled 30k reviewed UniRef sequences with length $\leq 350$. We then distributed these sampled sequences into training, test, and validation sets following a 80-10-10 split. The training data for UniRef$_L$ was curated by splitting UniRef50 data into buckets based on the distribution of lengths in our Cas9 dataset. We sampled sequences from each buckets such that the resulting set was as large as possible while still maintaining the Cas9 dataset's length distribution. We used MMSEQS to reduce redundancy of this set until the total dataset size was ∼50k, before distributed sequences into training, test, and validation sets following a 80-10-10 split.

**GFP Dataset.**    We curate our GFP dataset from two sources: FPBase (an online dataset of fluorescent proteins) and UniRef. For FPBase, we extract all basic proteins with an emission max within a desired range. From UniRef, we extract all sequences in the PF01353 PFam family. We then use MMSEQS to remove redundancy from the combined set, and distribute sequences into training, test, and validation sets following a 80-10-10 split.

**Cas9 Dataset.**    We curate a Cas9-family protein dataset from the CRISPR-Cas Atlas (Ruffolo et al., 2024), a large, publicly available resource that aggregates CRISPR-Cas systems and associated effector protein sequences across diverse microbial genomes. Starting from the full Atlas release, we first filter entries to retain only sequences annotated as Cas9 (discarding other Cas effectors and non-Cas proteins). This yields an initial pool of roughly $10^5$ candidate Cas9 sequences.

To reduce redundancy and obtain a compact but representative training set, we apply MMseqs2 redundancy reduction. Specifically, we cluster sequences by similarity and retain a representative sequence per cluster, reducing the dataset to around $2 \times 10^4$ diverse Cas9 sequences while preserving broad family coverage.

Finally, to prevent information leakage across splits, we assign train/validation/test sets at the cluster level rather than at the individual-sequence level. We perform a final clustering pass and then allocate entire clusters into the training, validation, and test splits using an 80-10-10 partition. This ensures that highly similar sequences do not appear in multiple splits, making generalization metrics more meaningful.

**Peptidomimetics Dataset.**    For training peptidomimetics Edit Flow, we curated a peptidomimetic sequence dataset by combining two sources: a commercial peptidomimetic library from ChemDiv and a manually expanded set generated from peptide SMILES using an RDKit-based transformation pipeline. For each input peptide, our script constructs candidate peptidomimetics through chemically motivated edits, including peptoid conversion, triazole isostere replacement, and for non cyclic peptides, N terminal and C terminal residue deletion, as well as a combined peptoid plus isostere edit. We further filter the generated peptidomimetics using synthetic accessibility and SCScore thresholds to remove implausible chemistries, and we remove exact duplicates to keep a clean dataset. Finally, we transform the remaining SMILES strings into SELFIES strings and use these SELFIES sequences to train the peptidomimetics Edit Flow.

## E.2    MODEL ARCHITECTURE

The backbone of our Edit Flow is a Transformer encoder that parameterizes a continuous time Markov chain over discrete edit operations. Input token sequences are first mapped to continuous vectors using a learned token embedding layer, and the normalized diffusion time is embedded with a time embedding module. Except for the peptidomimetic Edit Flow that applies a learned token embedding

layer, other Edit Flow apply a pre-trained ESM-2-650M model to compute the sequence embeddings (Lin et al., 2023). These two embeddings are combined and processed by a stack of Transformer blocks equipped with multi-head self attention, enabling the model to capture long range dependencies along the sequence. We use rotary positional embeddings to provide length generalization while avoiding fixed absolute position tables. Residual connections and layer normalization stabilize optimization, and a final layer normalization produces the hidden representation used by multiple output heads.

To model edit dynamics, the network predicts both the total edit rate and the distribution over edit types, using a reparameterization that separates the scalar intensity from the categorical choice among insertion, deletion, and substitution. Specifically, one head outputs a nonnegative total rate through a Softplus transformation, while a second head produces logits over the three edit types. Conditioned on the chosen type, token level heads output vocabulary logits for insertion and substitution, which define the token proposals at each position. This factorized design yields a flexible, stable parameterization of per position edit intensities and token proposals, and supports variable length sequence generation through insertion and deletion operations.

### E.3 TRAINING STRATEGY

During Edit Flow training, we use the least optimal alignment objective to supervise the edit process with an alignment between the source and target token sequences. For each terminal sequence $x_1$ in the dataset, we construct an initial sequence $x_0$ by sampling tokens from the vocabulary, with the initial length drawn uniformly from $[0, 2|x_1|]$ for the Peptidomimetic Edit Flow and $[(1 - a)|x_1|, (1 + a)|x_1|]$ for the protein Edit Flows. For UniRef$_S$, $a = 0.5$ and for UniRef$_L$ and Cas9 $a = 0.25$. This randomized initialization exposes the model to a broad range of starting lengths and content, encouraging robust insertion and deletion behavior and avoiding over-reliance on near identity alignments. Given the sampled $x_0$ and target $x_1$, we compute the least optimal alignment and train the model to match the induced edit dynamics, including both edit type intensities and token proposal distributions, along the continuous time editing trajectory.

**UniRef and Cas9 Edit Flow models.** Training was conducted on one NVIDIA RTX A6000 GPU with 48 GB of VRAM. The UniRef$_S$, UniRef$_L$, and Cas9 Edit Flow models were trained for around 30, 10, and 100 epochs respectively using the AdamW optimizer. We use a learning rate of $1 \times 10^{-5}$ and weight decay of 0.03. A learning-rate scheduler with 10 warm-up epochs and cosine decay is applied. We use the same warmup and minimum learning-rate setting as in our other Edit Flow trainings, with both the initial and minimum learning rates set to $1 \times 10^{-5}/10 = 1 \times 10^{-6}$. The network architecture uses a model dimension of 768, 8 Transformer layers, and 12 attention heads.

**GFP Edit Flow model.** Training was conducted on one NVIDIA RTX A6000 GPU with 48 GB of VRAM. We initialize the GFP Edit Flow from the pre-trained UniRef$_S$ checkpoint and fine-tune on the curated GFP dataset. The model is trained using the AdamW optimizer with weight decay 0.03. We use a learning rate of $1 \times 10^{-6}$ and apply a learning-rate scheduler with 10 warm-up epochs and cosine decay. As above, we set both the initial and minimum learning rates to $1 \times 10^{-6}/10 = 1 \times 10^{-7}$. The network architecture uses a model dimension of 768, 8 Transformer layers, and 12 attention heads.

**Peptidomimetic Edit Flow.** Training was conducted on one H100 NVIDIA NVL GPU with 94 GB of VRAM. The model was trained for 100 epochs using the AdamW optimizer and a learning rate of 3e-4 with weight decay of 0.03. A learning rate scheduler with 10 warm-up epochs and cosine decay was used, with initial and minimum learning rates both 3e-5. The network architecture included a model dimension of 768, 8 transformer layers, and 12 attention heads.

**Diversity Evaluation.** To quantify sample diversity for each Edit Flow, we draw 1000 sequences from the model and report an alignment-free $k$-mer Jaccard diversity computed on the generated sequences. For each sequence $s$, we form the set of all contiguous $k$-grams (we use $k = 3$) and estimate the average Jaccard similarity over a large number of randomly sampled sequence pairs. The reported diversity is defined as one minus this average similarity, $\text{Div}_k = 1 - \mathbb{E}[\text{Jaccard}(K_k(s_i), K_k(s_j))]$, which is robust to variable-length sequences and captures lexical diversity without requiring sequence alignment.

## F    SCORE MODEL DETAILS

### F.1    CAS9 SCORE MODELS

**Cas9 Classifier.**    We train a binary Cas9 classifier to predict whether a candidate protein sequence is a valid Cas9, and use this score both as an objective signal and as a quality-control component in Cas9 shrinkage experiments. The classifier is a lightweight MLP on top of a frozen ESM-2-650M sequence encoder: given an amino-acid sequence, we compute per-token embeddings with ESM-2-650M, apply mean pooling over non-padding tokens to obtain a fixed-length sequence representation, and then pass this representation through a small multi-layer perceptron to output a single logit (converted to a probability with a sigmoid).

We construct the positive set by sampling from the same curated Cas9 sequence corpus used to train the Cas9 Edit Flow. The negative set is designed to be challenging and length-matched: it includes (i) UniRef protein sequences sampled to have a similar length distribution to Cas9s, (ii) non-Cas9 Cas effector sequences from the CRISPR-Cas Atlas (e.g., other Cas families), and (iii) broken or corrupted Cas9-like sequences (e.g., truncated or otherwise implausible Cas9 sequences) to explicitly teach the classifier to reject malformed Cas9 candidates. Training was conducted on one NVIDIA RTX A6000 GPU with 48 GB of VRAM.

**PAM prediction.**    For our PAM prediction, we use the full-length protein2PAM model, available at 'Profluent-Bio/protein2pam-cas9_full'. Given that Protein2PAM outputs a distribution over predicted PAMs, enforcing exact PAM matching requires setting a threshold to determine what level of model confidence constitutes a nucleotide prediction vs. 'N'. Throughout all experiments, we set this threshold to 0.7, ensuring that only PAM predictions for which Protein2PAM is relatively confident are considered the 'correct PAM' for pCoMole to match.

### F.2    GFP SCORE MODELS

**GFP Classifier.**    We train a binary GFP classifier to enforce that generated sequences remain in the GFP family during GFP shrinkage experiments. The classifier is trained on the same curated GFP dataset used for GFP Edit Flows training. Architecturally, we encode each input sequence with a frozen ESM-2-650M encoder and apply mean pooling over non-special, non-padding tokens, followed by a small MLP head with two hidden layers and dropout to produce a single logit. We use stratified train/validation/test splits (0.70/0.15/0.15) and optimize a class-balanced binary cross-entropy loss with a positive-class weight computed from the training split.

Training was conducted on one H100 NVIDIA NVL GPU system with 94 GB of VRAM. The model was trained for 20 epochs using the AdamW optimizer and a learning rate of 2e-4 with weight decay of 0.01. A learning rate scheduler with 2 warm-up epochs and cosine decay was used, with initial and minimum learning rates both 2e-5. The network architecture included a model dimension of 512 and a 0.3 dropout rate. Training uses AdamW with a warmup-cosine learning-rate schedule and gradient clipping of 1.0. We select the best checkpoint by validation AUROC. On the held-out test set, the GFP classifier achieves loss 0.1834, AUROC 0.9784, AUPRC 0.9480, F1 0.9457, and accuracy 0.9506.

**FPredX.**    We use FPredX as our fluorescence predictor for GFP variants (Tam & Zhang, 2022). Given an amino acid sequence, FPredX outputs predicted brightness, excitation wavelength, and emission wavelength. We use these predictions to define the GFP objectives (maximize brightness and align excitation to the 488 nm laser line) and the emission-range constraint requiring the emission peak to fall within the green band. We use the pre-trained FPredX model without additional fine-tuning.

### F.3    PEPTIDOMIMETICS SCORE MODELS

**Peptiverse property predictors.**    For the peptidomimetic design tasks, we compute developability and binding-related objectives using pre-trained property prediction models from Peptiverse (Zhang et al., 2026). Given a candidate peptidomimetic sequence, these models output predicted non-toxicity, solubility, permeability, half-life, and binding affinity. We use these predicted scores directly as

objective values during pCoMole-guided sampling, and we apply the same fixed predictors across all methods and ablations to ensure a fair comparison.

**SMILES BindEvaluator.** To estimate pocket-level binding consistency for peptidomimetic editing, we train a SMILES BindEvaluator that predicts residue-wise binding sites given a binder SMILES sequence and a target protein amino-acid sequence. The model follows the same architecture as BindEvaluator (Chen et al.), but uses a pre-trained ChemBERTa-zinc250k-v2-40k model for tokenization and embedding pipeline for SMILES (Chithrananda et al., 2020), while the target sequence is encoded by pre-trained ESM-2-650M (Lin et al., 2023). The two representations are fused to produce per-residue binding-site probabilities. Training data are curated from PLINDER (Durairaj et al., 2024): we remove duplicates, filter target proteins with lengths in $[30, 800]$, merge binding-site annotations by majority vote for repeated (binder, target) pairs, drop entries with invalid SMILES or metals, and remove targets containing non-canonical amino acids. We further prevent leakage by removing validation examples that appear in the training set, and we augment the corpus with additional binder-target pairs from PepNN and BioLip2 after removing any overlap with PLINDER (Abdin et al., 2022; Zhang et al., 2024). After filtering, the final dataset contains 48155 training pairs and 6451 validation pairs.

Training was conducted on a 2xH100 NVIDIA NVL GPU system with 94 GB of VRAM. The model was trained for 50 epochs using the AdamW optimizer and a learning rate of 1e-4 with weight decay of 0.03. A learning rate scheduler with 5 warm-up epochs and cosine decay was used, with initial and minimum learning rates both 1e-5. The network architecture included a model dimension of 128, hidden dimension of 128, 8 transformer layers and a 0.3 dropout. During training, we tune the classification threshold on the validation set to maximize MCC, and we use the resulting threshold $\tau = 0.918$ for downstream scoring. SMILES BindEvaluator achieves a high prediction performance across all metrics.

**Motif and specificity scores.** Given a target of length $L$ (excluding special tokens) and SMILES BindEvaluator probabilities $p_i \in [0, 1]$ for each residue $i$ being in the binding site, we define two pocket-consistency scores. Let $\mathcal{M}$ denote the set of residues corresponding to the known binding motif/pocket region for the reference binder. The *motif score* is the mean predicted binding-site probability over motif residues,

$$\text{Motif}(x) \;=\; \frac{1}{|\mathcal{M}|} \sum_{i \in \mathcal{M}} p_i \,.$$

To quantify off-motif binding, we define *specificity* using the threshold $\tau$. we count the number of residues outside the motif whose predicted binding probability exceeds $\tau$, and normalize by sequence length,

$$\text{Specificity}(x) \;=\; 1 \;-\; \frac{1}{L} \sum_{i \notin \mathcal{M}} \mathbf{1}[p_i \geq \tau] \,.$$

Higher motif scores indicate stronger predicted binding in the intended pocket region, while higher specificity indicates fewer high-confidence binding-site predictions outside the known motif.

**Peptidomimetic Constraint.** We classify each SMILES as *natural peptide*, *peptidomimetic*, or *non-peptide/uncertain* using RDKit substructure matching and topology-based checks. We first detect amide (C=O-N) links and backbone "tiles" with SMARTS patterns for an $\alpha$-peptide segment (O=C-N-C(sp$^3$)-C=O) and for $\beta/\gamma$ spacing around an amide. A molecule is deemed $\alpha$-*peptide-only* if it contains at least two amide bonds, has at least one secondary amide (N-H), exhibits no $\beta/\gamma$ tiles, and its amides are largely covered by the $\alpha$-tile pattern (allowing up to two terminal amides). We then flag *peptidomimetic indicators* capturing deviations from a natural $\alpha$-peptide backbone, including backbone isosteres (depsipeptide ester links, thioamides, peptoid motifs), non-proline tertiary amides, and common linkage replacements such as ureas/thioureas, guanidines, sulfonamides, and imide/lactam/diacyl-amide motifs. If the $\alpha$-peptide-only criterion holds and no indicators are present we label the SMILES `natural_peptide`; if any indicator is detected we label it `peptidomimetic_not_natural`; otherwise we mark it `non_peptide_or_uncertain` and log an audit record of all SMARTS counts and triggered indicators.

**Length Constraint.** To determine whether a generated candidate is *shorter* or *smaller* than the starting molecule, we compute a backbone-length proxy and standard size descriptors with RD-Kit. We define *backbone centers* as amide carbonyl carbons (SMARTS `[CX3](=[OX1])[NX3]`) and optionally sulfonamide sulfur centers (SMARTS `[SX4](=[OX1])(=[OX1])[NX3]`), then build an undirected graph over centers by connecting two centers if there exists a single-bond path consistent with a peptide/peptidomimetic linkage: *center$_i$*-N-C(sp$^3$)-*center$_j$* (length 3) or *center$_i$*-N-(O/S/N)-C(sp$^3$)-*center$_j$* (length 4). We extract the largest connected component and use its number of nodes/edges (`backbone_centers_main`, `backbone_links_main`) as a proxy for backbone length, while also computing MW and heavy-atom count (HAC). A candidate is labeled `shorter_backbone` if its main-component nodes or edges decrease relative to the original, and `smaller_bulk` if both MW and HAC decrease (optionally by margins); we accept `overall_shorter_or_smaller` if either condition holds. During generation, we additionally enforce a heuristic token-length range $[L_0/2, L_0)$ to prevent extreme collapse while encouraging substantial compression.

## G  pCoMole Sampling Details.

**GFP Shrinkage.** For GFP shrinkage, unless otherwise stated, we use `num_steps=10`, `num_candidates=50`, and `num_rollouts=10`, with `objective_weights=[3, 1, 1]` for length reduction, brightness, and excitation alignment, respectively. To further bias sampling toward shorter sequences in this setting, we disable insertions by setting $\lambda_{\text{ins}} = 0$ and amplify deletions by multiplying the deletion rate by 100, which empirically encourages aggressive shrinkage while maintaining the GFP validity and emission constraints.

**Cas9 Shrinkage.** Across all Cas9 shrinkage experiments, we amplify the deletion rate by multiplying the base Edit Flow deletion intensity by 1000, which empirically encourages aggressive length reduction while still allowing the Doob-$h$ guidance to preserve Cas9-likeness and PAM constraints. Unless otherwise stated, we disable insertions by setting the insertion rate to zero (i.e., $\lambda_{\text{ins}} = 0$), since insertions are systematically misaligned with the shrinkage objective and can dilute search effort. Our PAM matching CE loss is comprised of two components: (i) the CE loss between the PAM logits and the target PAM nucleotide for all non-'N' positions; and (ii) the CE loss between the PAM logits and a uniform distribution. We use the PAM predicted by protein2PAM for the input sequence as the PAM to be maintained during pCoMole shrinkage.

*Main Cas9 results.* For the main Cas9 shrinkage experiments, we use `num_steps=20`, `num_candidates=10`, and `num_rollouts=5`. We set equal objective weights `objective_weights=[1,1,1]` (Cas9-likeness, PAM preservation loss, and length reduction, respectively), and keep insertions disabled via $\lambda_{\text{ins}} = 0$.

*Benchmark setting with fixed terminal length.* For the RayGun/SCISOR benchmark where we enforce an exact terminal length match, we use a shorter horizon but substantially increase lookahead to reduce variance under the hard length target: `num_steps=10`, `num_candidates=20`, and `num_rollouts=20`. We use `objective_weights=[1,3,1]` to place additional emphasis on PAM preservation during the more tightly constrained generation, and we again disable insertions ($\lambda_{\text{ins}} = 0$).

*Insertion ablation.* For the Cas9 insertion ablation, we match the main-results hyperparameters `num_steps=20`, `num_candidates=10`, `num_rollouts=5`, and `objective_weights=[1,1,1]`. The only change is whether insertions are permitted: in the *no-insertions* condition we set $\lambda_{\text{ins}} = 0$, while in the *insertions-enabled* condition we retain the base insertion rate from the pre-trained Edit Flow. In both conditions we keep the 1000× deletion-rate amplification.

**Peptidomimetic Binder Design.** Unless otherwise noted (e.g., in the hyperparameter ablations), we run pCoMole for peptidomimetic design with `num_steps=30`, `num_candidates=30`, and `num_rollout=10`. We use the fixed `objective_weights=[4, 2, 4, 0.2, 4, 2, 1]` for the seven peptidomimetic objectives respectively: non-toxicity, solubility, permeability, half-life, affinity, motif score, and specificity. Because the peptidomimetics Edit Flow backbone operates on SELFIES strings while all property and binding-score predictors take SMILES as input,

we convert each sampled SELFIES sequence to SMILES on the fly for objective and constraint evaluation at each sampling step. Reported average lengths (tokenized SMILES length) are rounded to integers for readability.

## H  BENCHMARK MODELS

**RayGun.**  For RayGun baselines, we use the publicly available pre-trained RayGun implementation with 8.8M parameters trained on UniRef50. All RayGun generations reported in this work are performed using this fixed pre-trained checkpoint. RayGun experiments were run on the H100 NVIDIA NVL GPU system with 94 GB of VRAM.

**SCISOR.**  For SCISOR baselines, we use the publicly available pre-trained `UniRef50_L` variant. We run SCISOR generation using this fixed pre-trained model without further fine-tuning. SCISOR experiments were run on one NVIDIA RTX A6000 GPU with 48 GB of VRAM.

## I  USE OF LARGE LANGUAGE MODELS (LLMS)

We acknowledge the use of large language models (LLMs) to assist in polishing and editing parts of this manuscript. LLMs were used to refine phrasing, improve clarity, and ensure consistency of style across sections. All technical content, experiments, analyses, and conclusions were developed by the authors, with LLM support limited to language refinement and editorial improvements.

Table S1: **Compute-matched best-of-$N$ comparison on GFP shrinkage.** Within the wall-clock time to generate one pCoMole design, SCISOR and RayGun generate 325 and 3655 candidates, respectively. We report best-of-$N$ scores up to these pool sizes. pCoMole reports five independent runs. All methods achieve near 100% validity rate. Best-of-$N$ candidates for SCISOR and RayGun are selected by the optical utility score $=$ brightness $- |\lambda_{\mathrm{exc}} - 488|$, consistent with the optical weights used in pCoMole. All methods enforce a fixed terminal length of 213 residues.

| Method | Best-of-N | $|\lambda_{\mathrm{exc}} - 488|$ ($\downarrow$) | Brightness |
|---|---|---|---|
| pCoMole | 1 | 3.8066 | 60.1518 |
| | 1 | 1.2353 | 76.4184 |
| | 1 | 2.5292 | 35.0746 |
| | 1 | 0.7497 | 32.0486 |
| | 1 | 5.7372 | 70.7304 |
| | Mean ± std (n=5) | 2.81 ± 2.02 | 54.88 ± 20.35 |
| SCISOR | 1 | 44.5227 | 20.6733 |
| | 10 | 36.5134 | 36.6634 |
| | 50 | 5.0683 | 34.5409 |
| | 100 | 8.4406 | 40.3289 |
| | 200 | 8.4406 | 40.3289 |
| | 334 | 8.4406 | 40.3289 |
| | All-candidate mean ± std | 38.73 ± 12.54 | 20.92 ± 6.70 |
| RayGun | 1 | 6.3094 | 15.6663 |
| | 50 | 6.2194 | 15.6723 |
| | 500 | 6.0357 | 16.6775 |
| | 1000 | 6.0357 | 16.6775 |
| | 2000 | 6.0357 | 16.6775 |
| | 3655 | 6.0357 | 16.6775 |
| | All-candidate mean ± std | 23.3262 ± 8.07 | 9.05 ± 3.13 |

Table S2: **Objective guidance ablation for pCoMole on GFP design.** For each objective guidance setting, we report the average metrics over 100 pCoMole-shrunk GFP sequences. Length reduction guidance is applied in all settings.

| Obejective Guidance $|\lambda_{\mathrm{exc}} - 488|$ | Brightness | $|\lambda_{\mathrm{exc}} - 488|$ ($\downarrow$) | Brightness | Length | Validity Rate |
|---|---|---|---|---|---|
| ✓ | ✓ | 3.7557 | 54.0159 | 214 | 1.0 |
| ✓ | ✗ | 2.9810 | 31.2600 | 213 | 1.0 |
| ✗ | ✓ | 12.9500 | 62.1200 | 214 | 1.0 |
| ✗ | ✗ | 32.8200 | 25.0200 | 213 | 1.0 |

Table S3: **Comparison of pCoMole to SCISOR and RayGun at shrinking CjCas9.** All three methods were used to shrink WT CjCas9 to a sequence length of 934.

| Method | Length Change | Avg. Cas9 Likelihood | PAM Match Rate |
|---|---|---|---|
| pCoMole w/ Cas9 Edit Flows | 984 $\longrightarrow$ 934 | 0.95 | 1.0 |
| pCoMole w/ UniRef Edit Flows | 984 $\longrightarrow$ 934 | 0.98 | 1.0 |
| SCISOR | 984 $\longrightarrow$ 934 | 0.82 | 0.35 |
| RayGun | 984 $\longrightarrow$ 934 | 0.0 | 0.0 |

Table S4: **Ablation of allowing pCoMole edit steps to make insertions.** In both cases, pCoMole was used to generate 50 shrunk variants of St1Cas9.

| pCoMole Config. | Length Change | Cas9 Likelihood | PAM Distr. CE | PAM Match Rate |
|---|---|---|---|---|
| pCoMole w/ ins. disabled | $1121 \rightarrow 1021$ $(-100)$ | $1.00 \rightarrow 0.98$ | $0.91 \rightarrow 0.89$ | 1.0 |
| pCoMole w/ ins. enabled | $1121 \rightarrow 1049$ $(-72)$ | $1.00 \rightarrow 0.97$ | $0.91 \rightarrow 0.90$ | 1.0 |

Table S5: **pCoMole designs short peptidomimetic binders for 15 PDB targets with known peptide binders, achieving substantial length reduction and improved predicted properties.** For each target, we report the average scores over 100 designed peptidomimetics and the property score change ($\Delta$) relative to the pre-existing binder. Length is the number of SMILES tokens after tokenization. All designed sequences satisfy the specified constraints, and positive $\Delta$ indicates improvement

| PDB | Length Change | $\Delta$Non-Toxicity | $\Delta$Solubility | $\Delta$Permeability | $\Delta$Half-life (h) | $\Delta$Affinity | $\Delta$Motif Score | $\Delta$Specificity |
|---|---|---|---|---|---|---|---|---|
| 1AYC | $59 \rightarrow 38$ | -0.1288 | +0.0163 | +0.1436 | +12.8558 | +0.1001 | +0.0268 | -0.0386 |
| 1B8Q | $51 \rightarrow 33$ | -0.2324 | +0.0456 | +0.1512 | +33.3370 | -0.0643 | +0.0024 | -0.0389 |
| 1DDV | $41 \rightarrow 28$ | +0.3973 | +0.0293 | +0.1945 | +31.1299 | +0.2025 | -0.0214 | +0.0098 |
| 1E6I | $43 \rightarrow 29$ | -0.1412 | +0.1025 | +0.1748 | +23.5607 | +0.1515 | +0.0547 | -0.0537 |
| 2LTV | $85 \rightarrow 64$ | +0.3128 | -0.0909 | +0.0584 | +9.6710 | +0.1178 | +0.1034 | -0.0978 |
| 2Q8Y | $72 \rightarrow 49$ | -0.0639 | -0.0591 | +0.0971 | +18.0023 | +0.2046 | +0.0122 | -0.0437 |
| 3IDJ | $50 \rightarrow 38$ | +0.1706 | -0.0250 | +0.2047 | +23.6463 | +0.0487 | -0.0953 | -0.0141 |
| 4GNE | $54 \rightarrow 39$ | -0.2138 | +0.2024 | +0.0570 | +17.7531 | +0.2295 | +0.0346 | -0.0570 |
| 5AZ8 | $80 \rightarrow 63$ | -0.2233 | -0.0964 | +0.0061 | +6.1258 | +0.0782 | +0.0934 | -0.0162 |
| 5KRI | $105 \rightarrow 87$ | -0.1807 | -0.0622 | +0.0079 | +2.6242 | +0.0738 | +0.0252 | -0.0448 |
| 5M02 | $61 \rightarrow 45$ | -0.2005 | -0.0956 | +0.0699 | +20.2843 | +0.2425 | +0.1078 | -0.0021 |
| 6MLC | $45 \rightarrow 30$ | -0.1460 | +0.0722 | +0.1717 | +20.0400 | +0.1731 | +0.0259 | -0.0053 |
| 7JVS | $79 \rightarrow 58$ | +0.0537 | -0.1033 | +0.0425 | +7.7421 | +0.1060 | +0.0240 | +0.0021 |
| 7LUL | $70 \rightarrow 47$ | +0.0936 | -0.0911 | +0.1835 | +23.2782 | +0.0769 | +0.0373 | -0.0320 |
| 8CN1 | $30 \rightarrow 23$ | +0.0239 | +0.0481 | +0.1976 | +31.8092 | +0.0371 | +0.0021 | -0.0416 |

Table S6: **Constraint ablation for peptidomimetic binder design on 3AMA and 5JHF.** For each target, we report the property scores of the known binder ("Baseline") and the average scores over 100 pCoMole-generated sequences under different constraint settings. Length is the number of tokenized SMILES tokens. PM denotes the hard peptidomimetic constraint and LEN denotes the soft length constraint. $R_{\text{PM}}$ and $R_{\text{LEN}}$ are the fractions (out of 100) of generated sequences that satisfy the PM and LEN constraints, respectively. Please refer to Section D for details.

| PDB | Constraints | Length | Non-Toxicity | Solubility | Permeability | Half-life (h) | Affinity | Motif Score | Specificity | $R_{\text{PM}}$ | $R_{\text{LEN}}$ |
|---|---|---|---|---|---|---|---|---|---|---|---|
| 3AMA | Baseline (Known Binder) | 141 | 0.901 | 0.854 | 0.2528 | 1.4508 | 0.6731 | 0.9171 | 0.9941 | — | — |
| | PM+LEN | 129 | 0.8085 | 0.4991 | 0.3049 | 4.8599 | 0.6946 | 0.9798 | 0.9429 | 1.00 | 1.00 |
| | PM only | 112 | 0.7444 | 0.5769 | 0.3537 | 14.4684 | 0.6666 | 0.8521 | 0.9482 | 1.00 | 0.27 |
| | LEN only | 131 | 0.8261 | 0.486 | 0.3081 | 4.9182 | 0.6947 | 0.9782 | 0.9443 | 1.00 | 1.00 |
| | None | 21 | 0.5104 | 0.8422 | 0.6186 | 68.4154 | 0.4696 | 0.4247 | 0.9613 | 0.10 | 0.02 |
| 5JHF | Baseline (Known Binder) | 95 | 0.8029 | 0.8744 | 0.2431 | 1.5936 | 0.5457 | 0.8484 | 0.9926 | — | — |
| | PM+LEN | 70 | 0.6989 | 0.6496 | 0.3296 | 9.2383 | 0.6327 | 0.9327 | 0.9576 | 1.00 | 1.00 |
| | PM only | 54 | 0.664 | 0.717 | 0.4032 | 19.8692 | 0.5861 | 0.7769 | 0.97 | 1.00 | 0.42 |
| | LEN only | 72 | 0.7131 | 0.6319 | 0.3342 | 8.6049 | 0.6319 | 0.9337 | 0.957 | 0.99 | 1.00 |
| | None | 7 | 0.3851 | 0.7816 | 0.7176 | 175.339 | 0.4342 | 0.5186 | 0.9997 | 0.00 | 0.00 |

Table S7: **Sampling-step ablation for peptidomimetic binder design on 2KXQ and 8EYA.** For each target, the row with #-steps= 0 reports the known binder (baseline), and the remaining rows report averages over 100 pCoMole-generated sequences under different numbers of sampling steps. Length is the number of tokenized SMILES tokens. We also report the average wall-clock time (seconds) to generate one sequence for each step setting.

| PDB | #-steps | Length | Non-Toxicity | Solubility | Permeability | Half-life (h) | Affinity | Motif Score | Specificity | Time (s) |
|---|---|---|---|---|---|---|---|---|---|---|
| | 0 | 130 | 0.5343 | 0.8607 | 0.2882 | 2.2298 | 0.6373 | 0.9528 | 0.8556 | — |
| | 10 | 116 | 0.8017 | 0.787 | 0.3000 | 3.5262 | 0.7565 | 0.9719 | 0.7786 | 69.73 |
| 2KXQ | 20 | 111 | 0.7683 | 0.7279 | 0.3113 | 4.4267 | 0.8055 | 0.9690 | 0.7882 | 150.94 |
| | 30 | 108 | 0.7626 | 0.7036 | 0.3180 | 5.3392 | 0.8106 | 0.9705 | 0.7837 | 257.37 |
| | 50 | 99 | 0.7541 | 0.6664 | 0.3330 | 6.7532 | 0.8037 | 0.9699 | 0.7928 | 530.83 |
| | 0 | 114 | 0.7220 | 0.8249 | 0.2410 | 4.2260 | 0.6285 | 0.9392 | 0.9967 | — |
| | 10 | 99 | 0.7137 | 0.7873 | 0.2848 | 4.6720 | 0.7010 | 0.9796 | 0.9840 | 67.33 |
| 8EYA | 20 | 96 | 0.6737 | 0.7670 | 0.3054 | 5.8047 | 0.7306 | 0.9796 | 0.9800 | 139.69 |
| | 30 | 93 | 0.6676 | 0.7459 | 0.3152 | 6.9622 | 0.7432 | 0.9791 | 0.9765 | 215.27 |
| | 50 | 84 | 0.6721 | 0.7146 | 0.3504 | 11.0632 | 0.7511 | 0.9788 | 0.9703 | 444.16 |

Table S8: **Rollout ablation for peptidomimetic binder design on 2KXQ and 8EYA.** For each target, the row with #-rollouts= 0 reports the known binder (baseline), and the remaining rows report averages over 100 pCoMole-generated sequences under different numbers of rollouts. Length is the number of tokenized SMILES tokens. We also report the average wall-clock time (seconds) to generate one sequence for each rollout setting.

| PDB | #-rollouts | Length | Non-Toxicity | Solubility | Permeability | Half-life (h) | Affinity | Motif Score | Specificity | Time (s) |
|---|---|---|---|---|---|---|---|---|---|---|
| | 0 | 111 | 0.8861 | 0.7156 | 0.2874 | 1.7210 | 0.6775 | 0.9323 | 0.9621 | — |
| | 5 | 87 | 0.7084 | 0.6840 | 0.2940 | 5.9379 | 0.7745 | 0.9371 | 0.9470 | 78.79 |
| 2QSC | 10 | 86 | 0.7172 | 0.6858 | 0.2974 | 6.2827 | 0.7809 | 0.9287 | 0.9477 | 124.57 |
| | 20 | 87 | 0.7447 | 0.6938 | 0.3063 | 6.7057 | 0.7826 | 0.9335 | 0.9464 | 216.92 |
| | 30 | 84 | 0.7175 | 0.6967 | 0.3042 | 7.1078 | 0.7756 | 0.9294 | 0.9462 | 306.83 |
| | 0 | 84 | 0.4621 | 0.7744 | 0.2787 | 2.5357 | 0.5218 | 0.3083 | 0.9954 | — |
| | 5 | 63 | 0.6625 | 0.7089 | 0.4353 | 15.4066 | 0.7062 | 0.6002 | 0.9914 | 131.07 |
| 7ZPY | 10 | 60 | 0.6886 | 0.7144 | 0.4490 | 17.4544 | 0.6988 | 0.6288 | 0.9936 | 205.87 |
| | 20 | 59 | 0.6694 | 0.7122 | 0.4435 | 18.1834 | 0.6940 | 0.6113 | 0.9920 | 344.24 |
| | 30 | 61 | 0.6729 | 0.7230 | 0.4505 | 19.0042 | 0.7020 | 0.616 | 0.9922 | 271.69 |

Table S9: **Number of candidates ablation for peptidomimetic binder design on 2KXQ and 8EYA.** For each target, the row with #-candidates= 0 reports the known binder (baseline), and the remaining rows report averages over 100 pCoMole-generated sequences under different numbers of candidates. Length is the number of tokenized SMILES tokens. We also report the average wall-clock time (seconds) to generate one sequence for each number of candidate setting.

| PDB | #-candidates | Length | Non-Toxicity | Solubility | Permeability | Half-life (h) | Affinity | Motif Score | Specificity | Time (s) |
|---|---|---|---|---|---|---|---|---|---|---|
| | 0 | 65 | 0.0826 | 0.8305 | 0.2977 | 2.7590 | 0.4128 | 0.9671 | 0.8806 | — |
| | 10 | 47 | 0.6998 | 0.7226 | 0.4626 | 16.5440 | 0.6143 | 0.9692 | 0.8327 | 41.42 |
| 2O9V | 20 | 45 | 0.7330 | 0.7345 | 0.4633 | 17.5760 | 0.6241 | 0.9646 | 0.8309 | 76.76 |
| | 30 | 46 | 0.7511 | 0.7323 | 0.4574 | 19.6885 | 0.6278 | 0.9658 | 0.8339 | 58.32 |
| | 50 | 47 | 0.7532 | 0.7379 | 0.4541 | 20.1591 | 0.6242 | 0.9683 | 0.8312 | 163.55 |
| | 0 | 143 | 0.6156 | 0.7538 | 0.185 | 6.1210 | 0.5750 | 0.6579 | 0.9502 | — |
| | 10 | 124 | 0.5925 | 0.7336 | 0.2478 | 6.2114 | 0.7314 | 0.7429 | 0.9300 | 113.14 |
| 8PFT | 20 | 120 | 0.5730 | 0.7208 | 0.2586 | 6.1821 | 0.7381 | 0.7548 | 0.9255 | 128.26 |
| | 30 | 115 | 0.5833 | 0.7183 | 0.2619 | 6.6555 | 0.7481 | 0.7498 | 0.9298 | 166.76 |
| | 50 | 116 | 0.5679 | 0.7094 | 0.2642 | 6.4031 | 0.7524 | 0.7631 | 0.9250 | 480.98 |

Table S10: **Pareto coverage comparison on PDB 4O56 under different scalarizations and weight vectors.** We evaluate three scalarization choices (ATC, Tchebycheff, and linear weighted sum) for multi-objective peptidomimetic binder design optimizing solubility, permeability, and binding affinity. For each method, we run pCoMole with three sets of objective weight vectors and report the average properties of 100 generated sequences. The rightmost column reports each method's empirical Pareto coverage against a reference front constructed by pooling solutions across all weight settings and methods (higher is better).

| Scalarization | Obejective Weights | | | Length | Solubility | Permeability | Affinity | Coverage v.s. Reference Front |
| | Solubility | Permeability | Affinity | | | | | |
|---|---|---|---|---|---|---|---|---|
| | 1 | 1 | 8 | 38 | 0.6856 | 0.3909 | 0.7488 | |
| ATC | 1 | 8 | 1 | 34 | 0.7131 | 0.4724 | 0.6832 | 0.7297 |
| | 8 | 1 | 1 | 37 | 0.8002 | 0.3851 | 0.6381 | |
| | 1 | 1 | 8 | 34 | 0.6973 | 0.4829 | 0.6748 | |
| Tchebycheff | 1 | 8 | 1 | 36 | 0.7248 | 0.4002 | 0.701 | 0.6486 |
| | 8 | 1 | 1 | 35 | 0.7197 | 0.4625 | 0.6509 | |
| | 1 | 1 | 8 | 38 | 0.7064 | 0.4052 | 0.7711 | |
| Weighted Sum | 1 | 8 | 1 | 34 | 0.7137 | 0.4866 | 0.6538 | 0.5946 |
| | 8 | 1 | 1 | 37 | 0.8051 | 0.3702 | 0.6348 | |

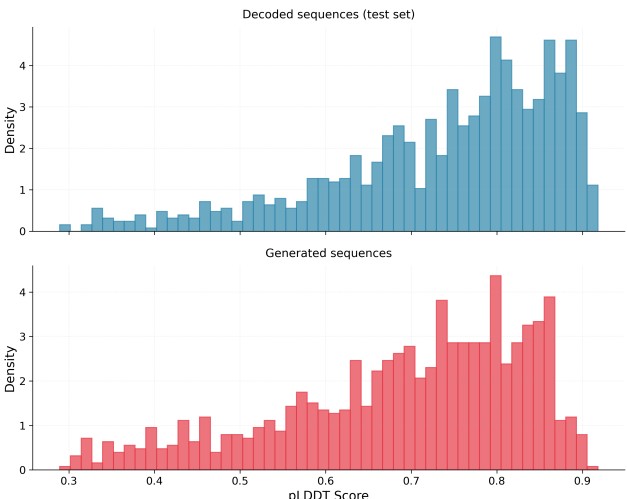

Figure S1: **Distribution of average pLDDT scores across set of 1000 sequences generated by the UniRef$_S$ Edit Flow, compared to the 1000 true Uniref sequences used as inputs.** 1000 UniRef sequences with length $\leq 350$ were sampled and used as $x_1$ for these generations.

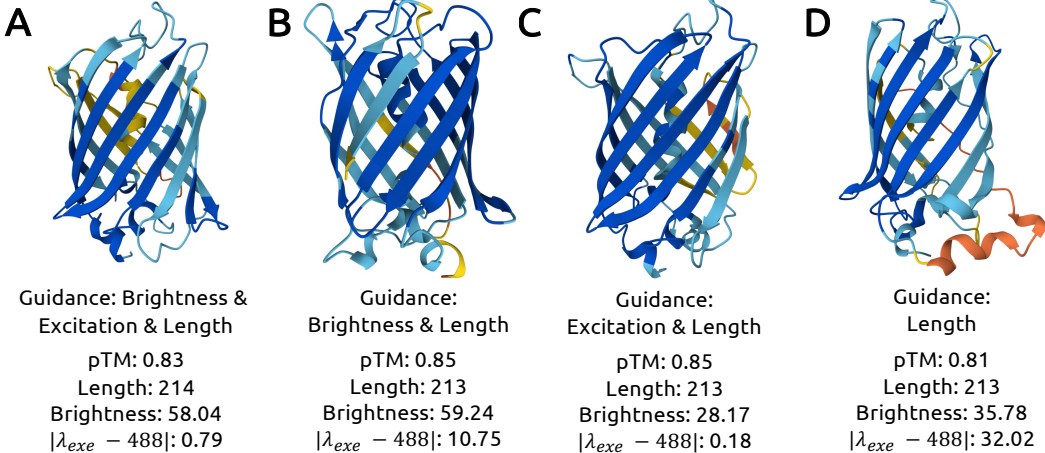

Figure S2: **Objective guidance controls optical trade-offs while preserving GFP structure under shrinkage.** Representative pCoMole-designed GFP variants generated under four objective-guidance settings: **(A)** brightness, excitation alignment, and length reduction **(B)** brightness and length reduction **(C)** excitation alignment and length reduction **(D)** length reduction only. We present their secondary structures together with AlphaFold3-predicted pTM scores, sequence length, predicted brightness, and excitation alignment error.

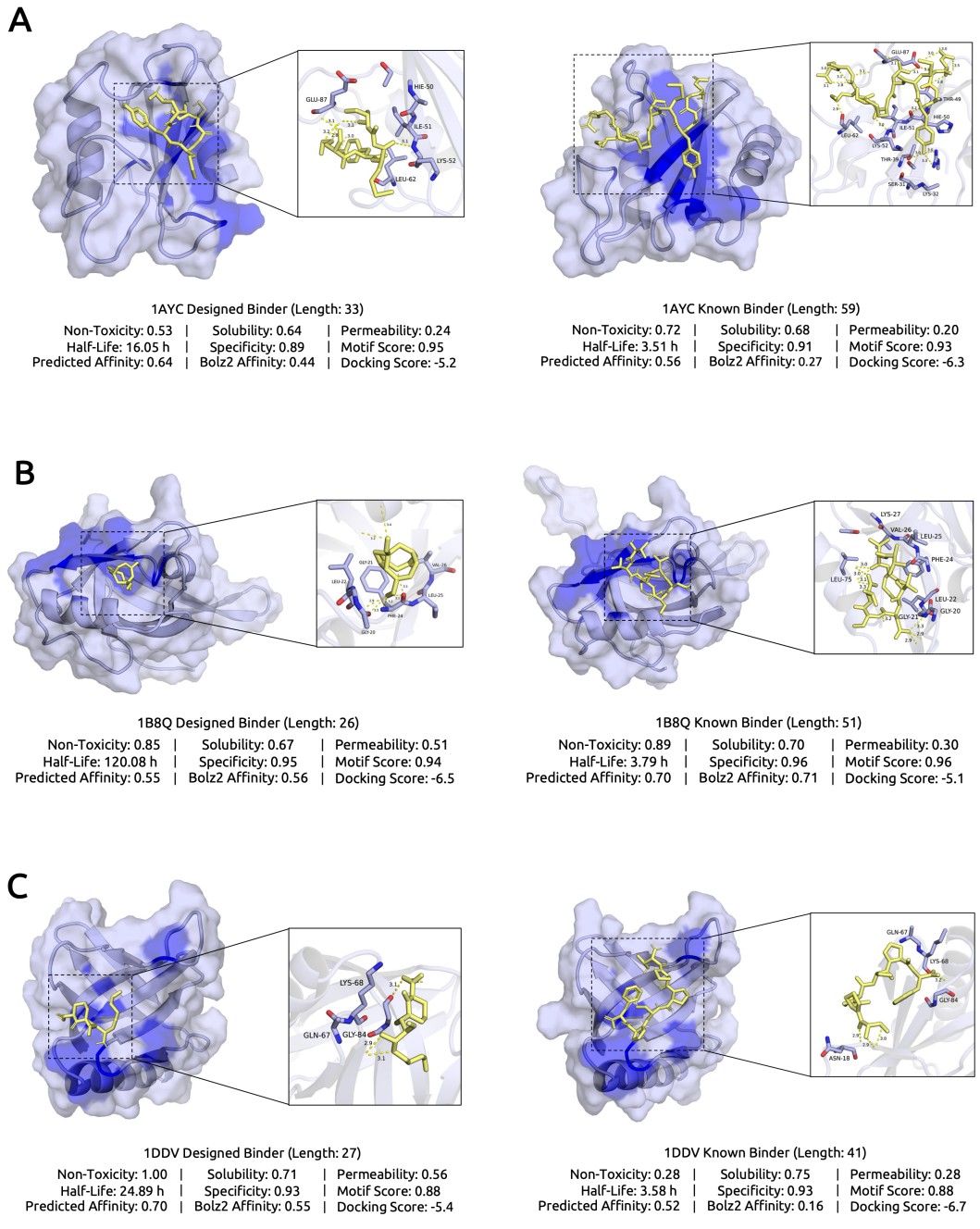

Figure S3: **Complex structures of known peptide binders and pCoMole-designed peptidomimetic binders with three target PDB proteins (A) 1AYC (B) 1B8Q (C) 1DDV.** Binders are shown in yellow, target proteins in light blue, and the dark-blue surface highlights the target motifs used by pCoMole to enforce motif-specific binding during design. Insets zoom into the binding interface to illustrate key contacts. Predicted property scores for each binder are reported.

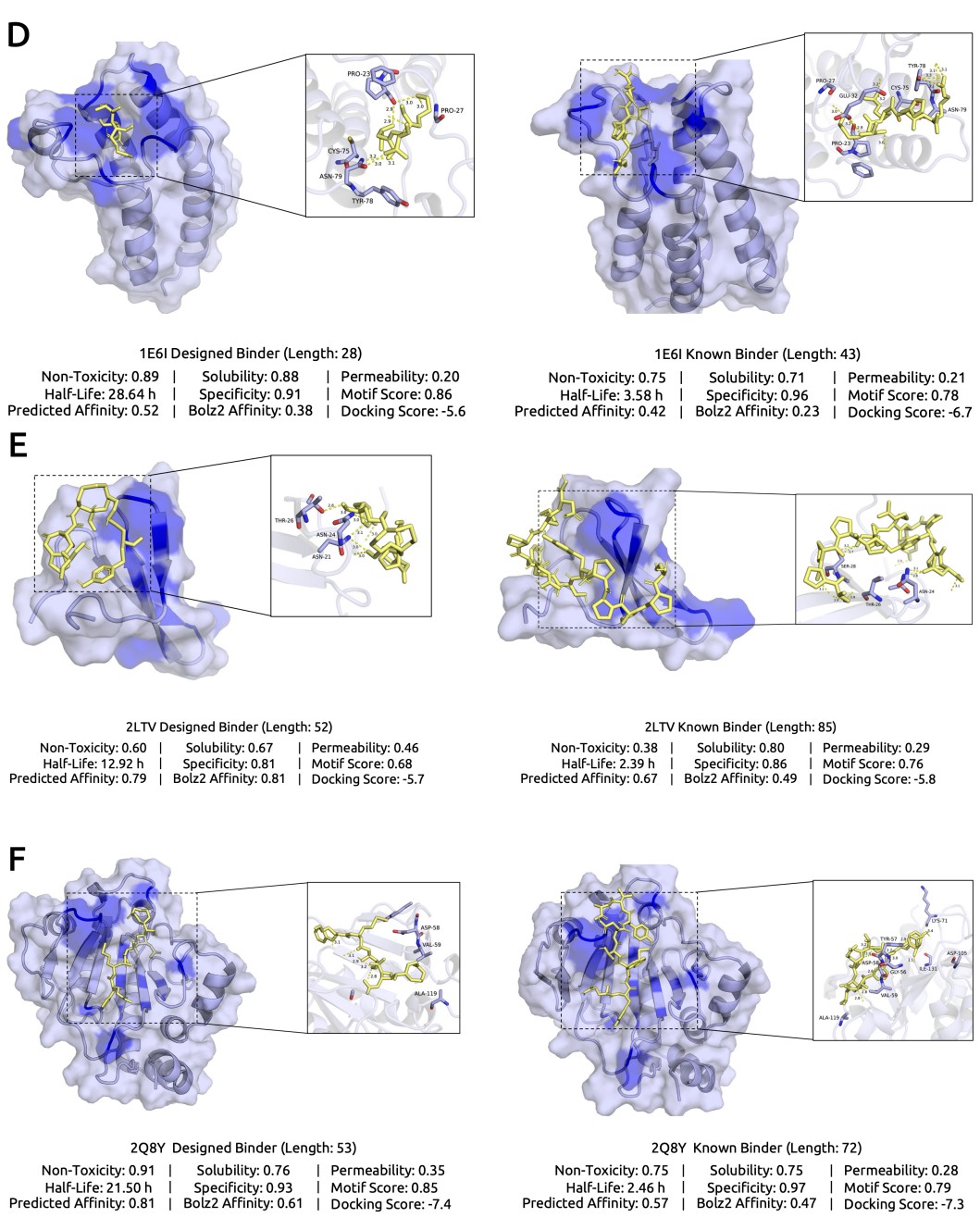

Figure S4: **Complex structures of known peptide binders and pCoMole-designed peptidomimetic binders with three target PDB proteins (D) 1E6I (E) 2LTV (F) 2B8Q.** Binders are shown in yellow, target proteins in light blue, and the dark-blue surface highlights the target motifs used by pCoMole to enforce motif-specific binding during design. Insets zoom into the binding interface to illustrate key contacts. Predicted property scores for each binder are reported.

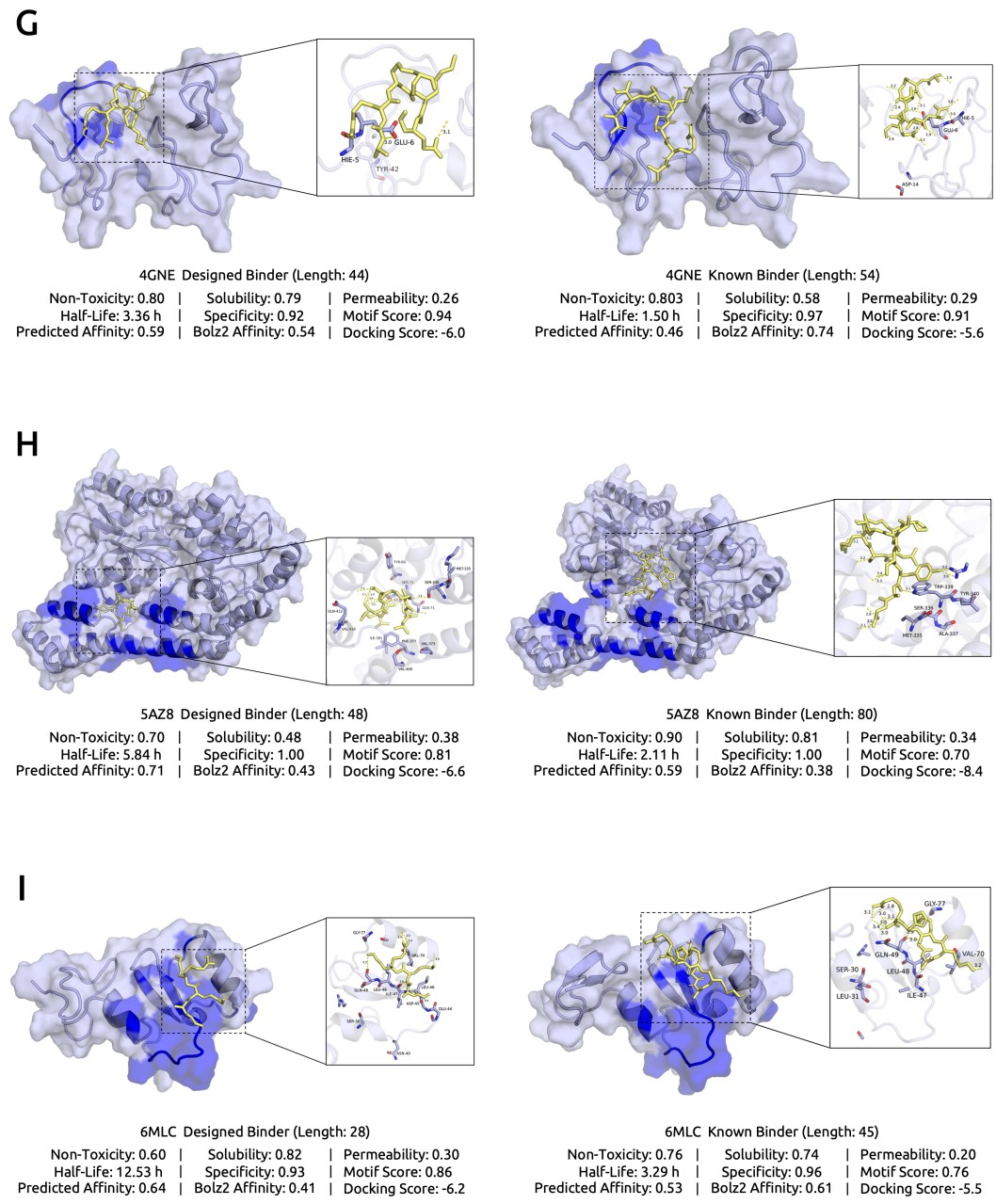

Figure S5: **Complex structures of known peptide binders and pCoMole-designed peptidomimetic binders with three target PDB proteins (G) 4GNE (H) 5AZ8 (I) 6MLC.** Binders are shown in yellow, target proteins in light blue, and the dark-blue surface highlights the target motifs used by pCoMole to enforce motif-specific binding during design. Insets zoom into the binding interface to illustrate key contacts. Predicted property scores for each binder are reported.

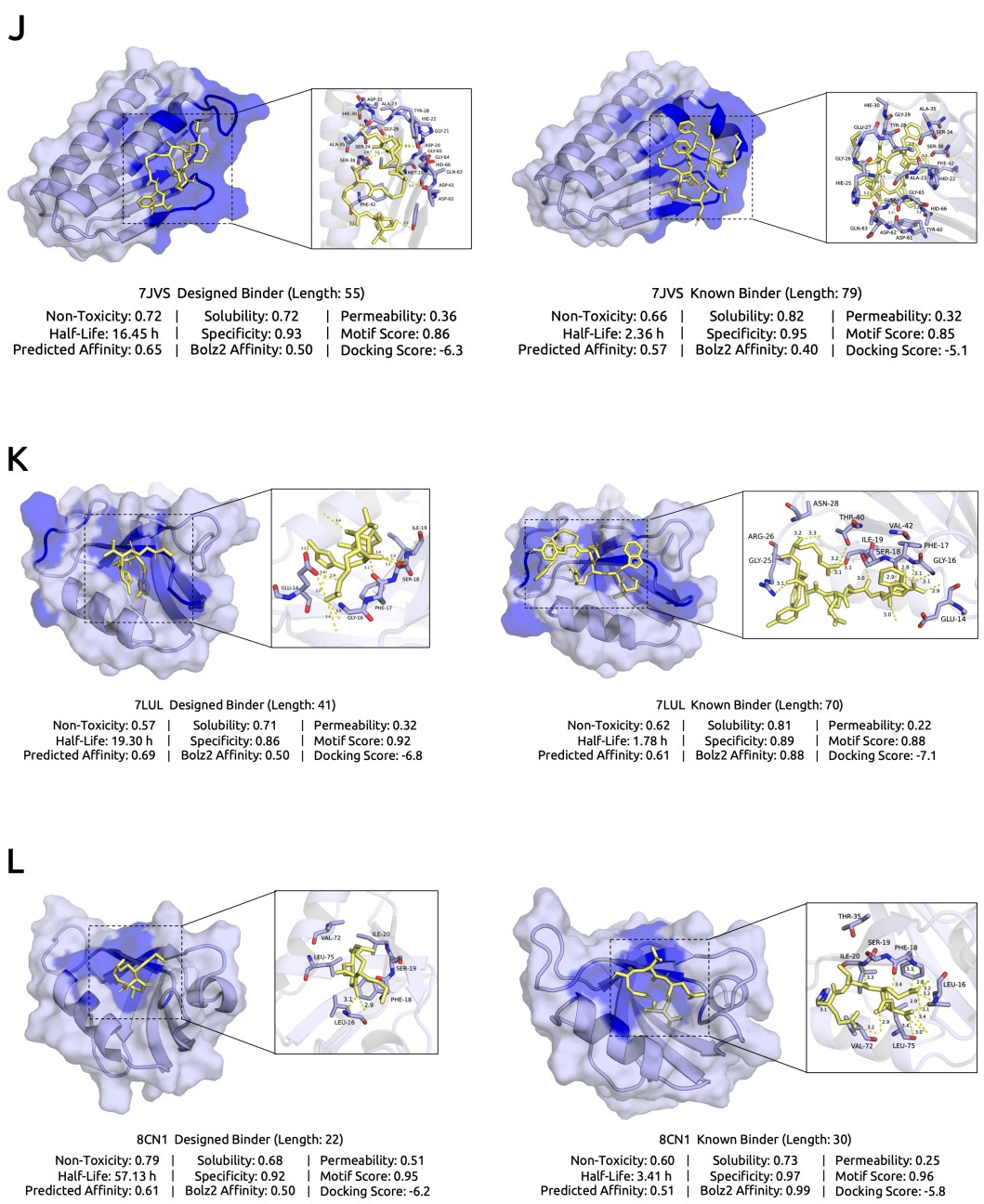

Figure S6: **Complex structures of known peptide binders and pCoMole-designed peptidomimetic binders with three target PDB proteins (J) 7JVS (K) 7LUL (L) 8CN1.** Binders are shown in yellow, target proteins in light blue, and the dark-blue surface highlights the target motifs used by pCoMole to enforce motif-specific binding during design. Insets zoom into the binding interface to illustrate key contacts. Predicted property scores for each binder are reported.

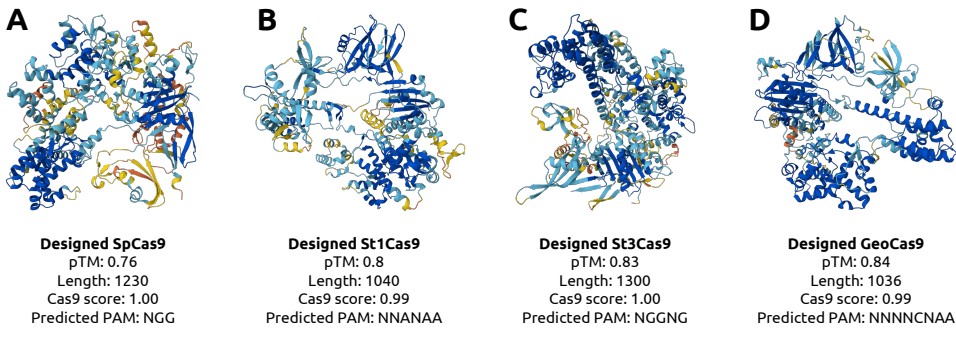

**Designed SpCas9**
pTM: 0.76
Length: 1230
Cas9 score: 1.00
Predicted PAM: NGG

**Designed St1Cas9**
pTM: 0.8
Length: 1040
Cas9 score: 0.99
Predicted PAM: NNANAA

**Designed St3Cas9**
pTM: 0.83
Length: 1300
Cas9 score: 1.00
Predicted PAM: NGGNG

**Designed GeoCas9**
pTM: 0.84
Length: 1036
Cas9 score: 0.99
Predicted PAM: NNNNCNAA

Figure S7: **pCoMole effectively shrinks diverse set of Cas9 orthologs.** Representative pCoMole-designed Cas9 sequences generated by shrinking four well-characterized Cas9 othologs: **(A)** SpCas9, **(B)** St1Cas9 **(C)** St3Cas9 **(D)** GeoCas9. We present their secondary structures together with AlphaFold3-predicted pTM scores, sequence length, Cas9-likeliness score, and predicted PAM.

