# OpenReview forum: "pCoMole: Pareto-Constrained Molecule Editing with Discrete Flows"
_ICLR.cc/2026/Workshop/LMRL — ICLR 2026 Workshop LMRL Poster_

### Official Review · Reviewer_Ny6y · 2026-02-17
**Review of pCoMole: Pareto-Constrained Molecule Editing with Discrete Flows**

**Rating:** 6
**Confidence:** 3

**Review:**

pCoMole presents a principled framework for constraint-aware, multi-objective biomolecular sequence editing built on discrete flow matching. The method steers pre-trained Edit Flows via a Doob-h transform guided by a feasibility-gated augmented utility, with provable preference consistency. Experiments across GFP shrinkage, Cas9 miniaturization, and peptidomimetic binder compression demonstrate consistent improvements over RayGun and SCISOR baselines.

The theoretical construction is rigorous and internally consistent, with a complete proof chain covering correctness, Pareto optimality, and approximation stability. Clarity is generally good, though a direct inconsistency between the text and Table 1 regarding train-test gaps undermines confidence in Section 4.1. While individual components are known, their combination in a unified offline editing framework with hard feasibility constraints and variable-length discrete dynamics is novel. Significance is promising but currently limited by the absence of wet-lab validation; all evaluations rely on pre-trained computational predictors.

## Strengths
- Complete theoretical chain from Doob-h correctness to multi-step stability guarantees
- Strong and consistent outperformance of RayGun and SCISOR on GFP and Cas9 tasks
- Broad experimental scope across proteins and small molecules

## Weaknesses
- All results are in silico, though the authors acknowledge this in Section 5
- Section 4.1 claims *"low train-test gaps"* but Table 1 shows a ~165% gap for the GFP Edit Flow (train 188.87 vs. test 501.58), which is neither discussed nor explained
- Pareto coverage analysis (Table S10) is limited to a single target and task, insufficient to support the general Pareto alignment claim

## Minor Issues
- Justification for absence of domain-specific baselines in Section 4.4 is only in Appendix A
- Typographical error in Introduction: unclosed parenthesis in "we introduce Pareto-Constrained Molecule editing (pCoMole a framework"

---

### Official Review · Reviewer_Sn46 · 2026-02-23

**Rating:** 8
**Confidence:** 3

**Review:**

The paper proposes the Pareto-Constrained Molecule Editing (pCoMole) framework with guardrails to create terminally feasible molecule through edits to existing molecules while adhering to user preferences. The framework is backed by theoretical contributions and is then tested on 3 different biomolecule editing tasks. The experiment results are encouraging as the method achieves shrinkage in length, improved values for the desired properties while retaining the intended function of the original biomolecule.

**Strengths:**
1) The proposed framework is technically sound. The idea of defining a terminal preference function and Doob-h transform and using it to prune exploration space is interesting.
2) The experiments are comprehensive and provide substantial evidence for the claims made by the authors.

**Weaknesses/Suggestions/Questions:**
1) **W:** The authors specify hard and soft constraints at several places in the experiments. However, it isn't clear from the theory or method description, if these constraints are treated differently when defining the feasible set?
2) **S:** Most experimental settings focus on shrinkage as one of the key objectives, which would imply that insertion is not particularly useful. It will be interesting to see a setting where insertion would be a useful operation as well.
3) **S:** There is no analysis of the distribution of number of times the different operations (insertion, deletion and substitution) are selected. It could give interesting insights into whether the method just prunes unnecessary components of the molecule or performs meaningful substitutions in the original sequence and if there's any pattern in those substitutions.
4) **S:** The work assumes access to accurate predictors of the properties, either through simulators or trained networks. It will be interesting to see how robust would the system be to inherent noise in the predictors?
5) **S:** There is often varying cost associated with obtaining different property values for a molecule. It could be worthwhile to investigate how to impose broader trajectory based constraints like limiting the number of calls to some expensive oracles.

---

### Official Review · Reviewer_Xz3B · 2026-02-24
**pCoMole: Pareto-Constrained Molecule Editing with Discrete Flows**

**Rating:** 7
**Confidence:** 4

**Review:**

This paper introduces pCoMole, a discrete flow-matching framework for multi-objective biomolecular editing under hard feasibility constraints. By combining an augmented Tchebycheff utility with a Doob-h transform, the method steers edit flows toward Pareto-aligned, constraint-satisfying solutions in variable-length biological sequences.
Strengths

Principled formulation integrating Pareto multi-objective optimization with hard feasibility constraints via Doob-h transform.
Handles discrete, variable-length sequence editing, which is challenging in biomolecular design.
Strong application demonstrations (GFP shrinking, Cas9 compression, peptide optimization with multiple drug-related constraints).

Weaknesses

Limited clarity on computational cost and scalability of Monte Carlo rollout approximation.
No extensive comparison against alternative multi-objective generative or constrained optimization baselines.
Feasibility guarantees depend on approximations; theoretical bounds on approximation error are not deeply analyzed.

---

### Meta-Review · Area_Chair_mYGM · 2026-02-27

**Recommendation:** Accept (Poster)
**Confidence:** 4

**Metareview:**

Accept.

---

### Decision · Program_Chairs · 2026-03-02

**Decision:**

Accept (Poster)

**Comment:**

Please see the meta-review.